# FROM MEDICAL RECORDS TO DIAGNOSTIC DIALOGUES: A CLINICAL-GROUNDED APPROACH AND DATASET FOR PSYCHIATRIC COMORBIDITY

**Tianxi Wan[1], Jiaming Luo[1], Siyuan Chen[1], Kunyao Lan[1], Jianhua Chen[2]**
**Haiyang Geng[3], Mengyue Wu[1]***

[1] X-LANCE Lab, Shanghai Jiao Tong University, China
[2] Shanghai Mental Health Center, SJTU School of Medicine, China
[3] Chen Frontier Lab for AI and Mental Health, Tianqiao and Chrissy Chen Institute, Shanghai, China
`{xixi0630, leojm2017, lankunyao, mengyuewu}@sjtu.edu.cn`
`chensiyuan925@outlook.com  jianhua.chen@smhc.org.cn`
`genghaiyang@cheninstitute.org`

## ABSTRACT

Psychiatric comorbidity is clinically significant yet challenging due to the complexity of multiple co-occurring disorders. To address this, we develop a novel approach integrating synthetic patient electronic medical record (EMR) construction and multi-agent diagnostic dialogue generation. We create 502 synthetic EMRs for common comorbid conditions using a pipeline that ensures clinical relevance and diversity. Our multi-agent framework transfers the clinical interview protocol into a hierarchical state machine and context tree, supporting over 130 diagnostic states while maintaining clinical standards. Through this rigorous process, we construct PsyCoTalk, the first large-scale dialogue dataset supporting comorbidity, containing 3,000 multi-turn diagnostic dialogues validated by psychiatrists. This dataset enhances diagnostic accuracy and treatment planning, offering a valuable resource for psychiatric comorbidity research. Compared to real-world clinical transcripts, PsyCoTalk exhibits high structural and linguistic fidelity in terms of dialogue length, token distribution, and diagnostic reasoning strategies. Licensed psychiatrists confirm the realism and diagnostic validity of the dialogues. This dataset enables the development and evaluation of models capable of multi-disorder psychiatric screening in a single conversational pass.

## 1 INTRODUCTION

Psychiatric disorders account for over 125 million disability-adjusted life years globally (Collaborators et al., 2022). A major challenge lies in psychiatric comorbidity, i.e., the co-occurrence of multiple conditions, which significantly complicates diagnosis and treatment. For instance, in a Netherlands study of depression and anxiety, 67% of individuals with a primary depression diagnosis had current and 75% had lifetime comorbid anxiety disorders (Lamers et al., 2011). Yet, most existing datasets and models narrowly focus on single disorders (Aich et al., 2024; Yao et al., 2022), while the few corpora covering multiple conditions lack fine-grained annotations and fail to capture symptom co-occurrence and progression within diagnostic processes (Sun et al., 2021; Yin et al., 2025; Cohan et al., 2018). As a result, large language models (LLMs) have not been systematically evaluated on multi-disorder diagnostic tasks, limiting the development of reliable screening systems that require step-by-step reasoning grounded in DSM-5 standards (Association et al., 2013). [1]

To construct large-scale comorbidity diagnostic dialogues, diverse patient profiles reflecting real-world complexity are needed. Such dialogues are essential because they simulate step-by-step di-

---

*Corresponding author.

[1] DSM-5 is the fifth edition of the Diagnostic and Statistical Manual of Mental Disorders by the American Psychiatric Association. It provides standardized criteria for diagnosing mental disorders and is used to ensure accuracy and consistency.

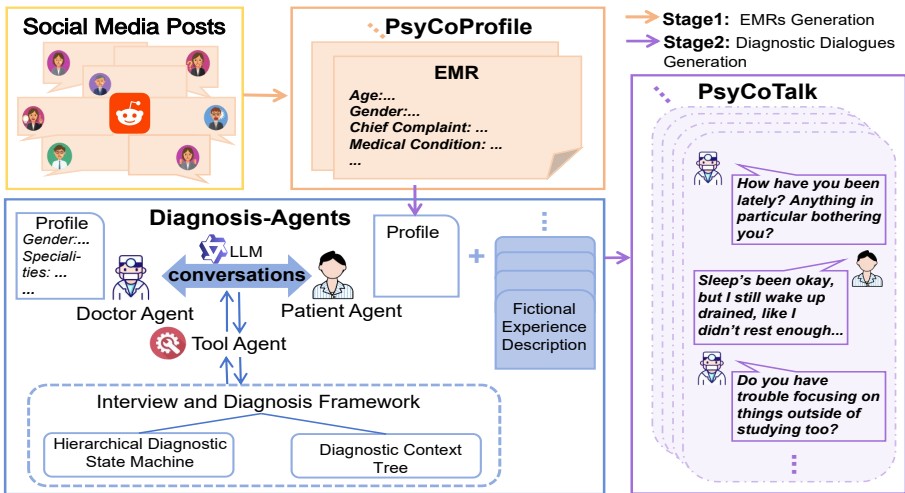

Figure 1: Framework overview: 1) **EMR**: construct patient profiles with 6 comorbidity types in electronic medical record structure by extracting social media posts of users self-reporting multiple disorders; 2) **Dialogue**: build a multi-agent framework with hierarchical state machines, based on SCID-5-RV (First et al., 2014), a standardized and semi-structured interview guide for major disorders assessment, to construct a comorbidity-focused diagnostic dialogue dataset **PsyCoTalk**.

agnostic reasoning and enable downstream applications such as multi-disorder screening, clinical decision support, and the training of dialogue agents for psychiatry. Yet profiles alone often lack structured detail to support accurate diagnosis. Electronic medical records (EMRs), by contrast, provide a standardized format that includes a wide range of clinical data, such as symptoms and medical history, which are crucial for later training. Still, EMRs can not handle diverse and complex clinical scenarios during diagnosis and treatment planning and do not capture the dynamic doctor–patient interaction, making it necessary to design a structured clinical dialogue flow that guides the diagnostic process and ensures realism and clinical validity.

This work aims to advance comorbidity diagnosis through the construction of PsyCoTalk, a large-scale dialogue dataset designed for both data-driven modeling and systematic evaluation of diagnostic reasoning. Our framework transforms self-reported social media posts into structured EMRs, which serve directly as patient agent profiles to drive a three-agent diagnostic system (doctor, patient, tool) for generating clinically grounded multi-turn dialogues. As illustrated in Figure 1, we introduce an innovative data construction and dialogue generation pipeline and make three key contributions:

• We develop a nuanced pipeline for EMR-driven dialogue generation that constructs synthetic, clinically grounded EMRs for comorbidity patients, offering a set of 502 synthetic EMRs along with detailed personal experiences reflecting common comorbid conditions, including *Depression, Anxiety, Bipolar and Attention-Deficits*.

• We propose a clinically grounded multi-agent framework combining a Hierarchical Diagnostic State Machine (HDSM) and Diagnosis Context Tree (DCT) for diagnosis, inspired by clinical interview manuals and covering 130+ diagnostic states while following psychiatric assessment protocols.

• We introduce **PsyCoTalk**, a large-scale dataset of 3,000 multi-turn diagnostic dialogues grounded in our EMRs and validated by psychiatrists, which is the largest of its kind and features longer and more clinically deep dialogues compared to existing corpora.

## 2 RELATED WORK

**Mental-health dialogue corpora**    Recent datasets fall into two groups. *Single-disorder resources* dominate: $D^4$ (Yao et al., 2022) contains 1 339 Chinese doctor–patient dialogues on depression; PsyQA (Sun et al., 2021) offers 22 000 question–answer pairs, later enlarged to 55 000 supportive

conversations in SMILECHAT (Qiu et al., 2023); and EFAQA (Hai Liang Wang, 2020) adds 20 000 real counselling sessions. *Broader clinical sets* remain rare. CED-BS (Aich et al., 2024) targets bipolar disorder and schizophrenia, while MDD-5k (Yin et al., 2025) is the largest Chinese diagnostic corpus so far, with 5 000 multi-turn simulations covering more than 25 disorders. Although MDD-5k uses a one-to-many case-to-dialogue strategy to boost diversity, it still treats each illness in isolation and offers limited control over combinatorial symptom paths.

**LLM-driven dialogue simulation** LLMs now power many mental-health studies. Early work assessed general chatbots: LLM-Empowered Chatbots (Chen et al., 2023) measured ChatGPT's diagnostic empathy, and Patient-Ψ (Wang et al., 2024) combined cognitive-behavioural-therapy rules with LLMs to tutor counsellors. CPsyCoun (Zhang et al., 2024) turned structured therapy notes into trainable conversations, improving topic coverage. Multi-agent designs add stronger clinical grounding. The AMC framework (Lan et al., 2024) links doctor, patient, and supervisor agents through a three-level memory, enabling domain adaptation without fine-tuning. Building on AMC, MDD-5k introduced a neuro-symbolic controller and dynamic diagnosis tree to steer topic flow (Yin et al., 2025). These ideas inform our **HDSM-Agents**, which extends the agent paradigm with a hierarchical state machine and context tree explicitly aligned to authoritative interview standards and, for the first time, generates dialogues reflecting comorbid diagnostic reasoning. Yet no prior corpus or simulator provides a large-scale, clinically structured resource for psychiatric comorbidity. Our work addresses this critical gap for the first time.

## 3    PATIENT EMR AND EXPERIENCE GENERATION WORKFLOW

To create standardized EMRs for training purposes, we collaborated with psychiatrists to develop a reference standard based on real clinical cases. Each EMR includes seven key components: *Demographic Information*, *Chief Complaint*, *Medical Condition*, *Medical History*, *Personal History*, *Family History*, and *Preliminary Diagnosis*. Mental status examination and auxiliary tests are excluded due to the limitations of social media data. Detailed descriptions of the content structure and language requirements for each section are provided in Appendix A.3.

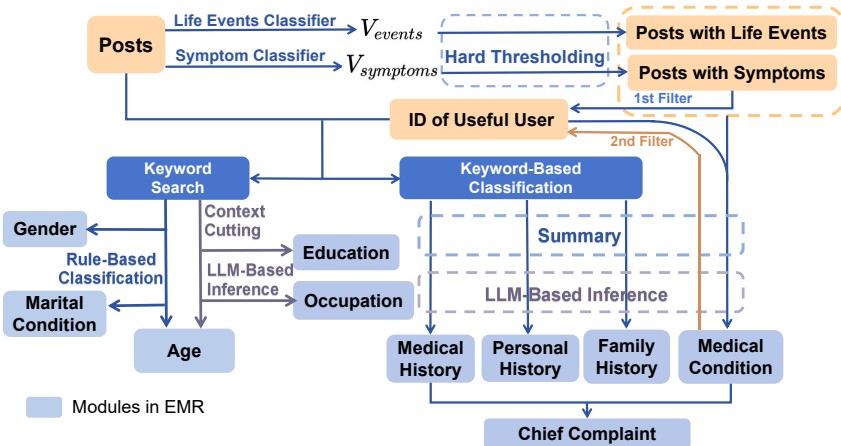

Figure 2: Overview of the EMR generation pipeline. Starting from social media posts, different modules of the EMR are generated using distinct methods, with the overall process proceeding in a top-down manner. The LLM used for EMR generation is GPT-4o-mini (Achiam et al., 2023).

### 3.1    EMR GENERATION

**Dataset Selection** We selected the PsySym (Zhang et al., 2022) dataset as our base due to its scale, depth of annotation, and suitability for modeling psychiatric comorbidity. PsySym originates from Reddit users with self-reported psychiatric disorders, offering fine-grained symptom-level annotations across 7 major mental disorders. The dataset contains 5,624 diagnosed users, with an average of 102.5 posts per user, making it ideal for our needs.

**User Filtering** To ensure meaningful interactions, we filtered users based on symptom diversity

and frequency. We retained users with at least 10 symptom-related posts and 20 distinct symptom types. Using a DSM-5-aligned disease–symptom graph, we excluded users with inconsistent symptom–label pairs, ensuring clinical reliability.

**Generation Strategy** We adopted a modular approach to EMR generation, which outperformed a single-step aggregation method in terms of information recall, classification accuracy, and reasoning coherence. This approach involves categorizing post content into the seven standard EMR sections, generating each section individually using either rule-based methods or LLM inference, and merging them into a complete record. During EMR generation, we iteratively sampled cases for expert review by eight psychiatrists; their feedback guided revisions to ensure medical validity.

**Processing Strategies** As illustrated in Figure 2, we designed four strategies for information extraction and generation:

- **Chief Complaint and Medical Condition:** Using dual classifiers for symptoms and life events, we generate binary symptom-event vectors for each post, which are then summarized and integrated into prompts for LLM-based generation.

- **Medical, Personal, and Family History:** Posts are categorized using keyword-based classification, summarized, and grouped into segments for LLM inference.

- **Education, Occupation, and Implicit Age:** Relevant content is retrieved through keyword search, trimmed with local context, and formatted into structured prompts.

- **Gender, Marital Status, and Explicit Age:** Demographic information is extracted directly using predefined keyword rules and rule-based classification.

## 3.2 PSYCOPROFILE DATASET DESCRIPTION AND STATISTICS

After two rounds of user filtering and using a modular strategy for content extraction and label injection, we generated 502 structured EMRs from retained texts. These cover six disease combinations of four core psychiatric conditions: *Major Depressive Disorder (MDD), Anxiety Disorder (AD), Bipolar Disorder (BD), and Attention-Deficit/Hyperactivity Disorder (ADHD)*. Table 1 shows user counts before and after filtering, plus statistics on average posts, posts with symptoms, life events, and distinct symptoms. Results show retained users provide rich, dense, analyzable content.

Table 1: **User filtering and posting statistics by disease combination.** *U-1st, U-2nd*: users after first and second filtering; *PS*: symptom posts; *PLE*: life events; *ST*: symptoms.

| DC | Initial | U-1st | U-2nd | Posts | PS | PLE | ST |
|---|---|---|---|---|---|---|---|
| ['AD', 'MDD'] | 310 | 157 | 141 | 127 | 25 | 13 | 27 |
| ['BD', 'MDD'] | 237 | 124 | 66 | 126 | 25 | 14 | 27 |
| ['ADHD', 'AD', 'MDD'] | 194 | 99 | 92 | 110 | 23 | 12 | 27 |
| ['ADHD', 'MDD'] | 220 | 88 | 75 | 167 | 22 | 11 | 27 |
| ['AD', 'BD', 'MDD'] | 106 | 74 | 73 | 110 | 34 | 19 | 28 |
| ['ADHD', 'AD'] | 146 | 69 | 55 | 163 | 23 | 11 | 27 |
| **Average** | 184 | 95 | 75 | 134 | 25 | 13 | 27 |

To validate the realism of the synthetic EMRs, we compared them with real-world data[2] across disease distribution, demographic factors (age and gender), and family history. The results are illustrated in Figure 3. Disease prevalence in synthetic EMRs broadly aligns with real-world data while exhibiting a more balanced distribution. Age shows a peak in the 20–24 group for synthetic data versus 30–34 for real records (Collaborators et al., 2022), likely reflecting the younger social media demographic (DataReportal, 2020). Gender proportions are more balanced in synthetic records, consistent with reduced stigma in online contexts (Porteous & Armstrong, 2021). Family history is slightly overrepresented but remains within clinically plausible bounds. Psychiatrist feedback on sampled records confirmed sufficient diversity and clinical plausibility, closely matching real-world documentation.

---

[2]For real-world comparison, we obtained approximately 1,000 de-identified clinical records from the National Mental Health Center, provided with institutional approval and used strictly for research purposes.

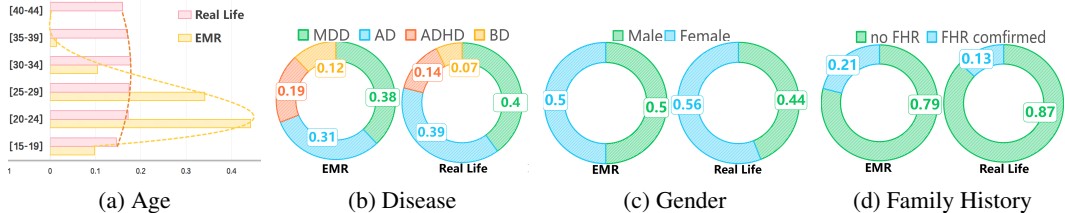

Figure 3: Comparisons between synthetic EMRs and real-world data.

### 3.3 FICTITIOUS PATIENT EXPERIENCE GENERATION

EMRs provide structured and factual information, but they lack the narrative richness and variability found in real patient-doctor interactions. By generating personalized fictitious experiences, we can create a more comprehensive and varied dataset that includes not only the factual data from EMRs but also the nuanced, context-specific details that arise in clinical conversations. Psychiatric diagnoses primarily rely on language-based interactions, making it feasible to generate multiple coherent diagnostic dialogues from a single EMR, provided the symptom presentation remains consistent with the diagnosis (Yin et al., 2025).

We therefore propose a personalized fictitious experience generation method grounded in structured EMRs to increase the diversity and realism of simulated dialogues. Unlike MDD-5k, which randomly samples templates, our approach aligns sampled content with EMR attributes to avoid semantic conflicts and incorporates clinically relevant lifestyle attributes. This ensures both semantic coherence and diagnostic validity in generated dialogues.

Concretely, we employ LLMs to process each structured EMR via prompt-based instruction tuning. For each case, the model is guided to generate a total of 5 personalized *Personal Histories*, including both existing entries from the EMR and newly generated ones when necessary. In addition, 10 semantically consistent *Fictitious Experiences* are produced, all aligned with the EMR content and free from logical conflicts. Here, *Personal Histories* refer to lifestyle or health background (e.g., *"prefers light food, smokes and drinks occasionally, and exercises three times per week"*), while *Fictitious Experiences* describe past events potentially affecting mental health (e.g., *"was blamed as the main cause of a failed company project one year ago"*). As shown in Figure 4, the generation process outputs two structured dictionaries: $\mathcal{D}_{his}$ for personal histories and $\mathcal{D}_{fic}$ for fictitious experiences. Given a structured EMR input $\mathbf{x}_{EMR}$ and $\tilde{e}$ represents the free-text narrative description corresponding to a selected fictitious experience $e$. Formally, we define:

$$\mathcal{D}_{fic}, \; \mathcal{D}_{his} = \text{LLM}\big(\text{Prompt}(\mathbf{x}_{EMR})\big)$$
$$\tilde{e} = \text{LLM}\big(\text{Prompt}(h, e)\big), \quad h \in \mathcal{D}_{his}, \; e \in \mathcal{D}_{fic} \tag{1}$$

This two-stage generation mechanism enables each EMR to yield up to 50 unique fictitious experiences, thereby enhancing the diversity, flexibility, and clinical realism of downstream diagnostic dialogues while preserving semantic consistency. For detailed description of prompt please refer to Appendix A.4.

## 4 MULTI-AGENT DIAGNOSIS FRAMEWORK

To construct a clinically realistic and diverse dataset for psychiatric diagnostic dialogues, we developed a multi-agent framework, that integrates structured EMRs with dynamic dialogue generation. This framework comprises two main components: a clinical-grounded interview pipeline, and a multi-agent execution mechanism.

### 4.1 CLINICAL-GROUNDED INTERVIEW PIPELINE

The Structured Clinical Interview for DSM-5 (SCID-5-RV) is a standardized tool for assessing mental disorders, offering a systematic approach to symptom evaluation and diagnosis. Its diagnostic

structure has been widely validated in diverse clinical and cultural settings (Mohammadkhani et al., 2020; Shankman et al., 2018; Brodey et al., 2018). Adhering to SCID-5-RV grounds our framework in established clinical practices. To guide LLM-based psychiatric interviews, we restructure SCID-5-RV into two parts: a Hierarchical Diagnostic State Machine (HDSM) and a Diagnostic Context Tree (DCT). This design ensures a structured, dynamic, and clinically coherent dialogue flow.

**Hierarchical Diagnostic State Machine (HDSM)** The HDSM adheres to the SCID-5-RV protocol, assigning one sub-state machine to each target disorder (MDD, AD, BD, ADHD). Each sub-state machine terminates in explicit diagnostic outcomes; for MDD, the terminal states `depression1–depression5` in Figure 5 represent five mutually exclusive diagnoses (see the complete architecture in Appendix A.7). In contrast to MDD-5k, which selected topics in random order, HDSM enables the agent to both ask questions and refine the diagnosis iteratively, reflecting real clinical reasoning. Following clinical design, the HDSM consists of a *Graphical three-level hierarchy:*

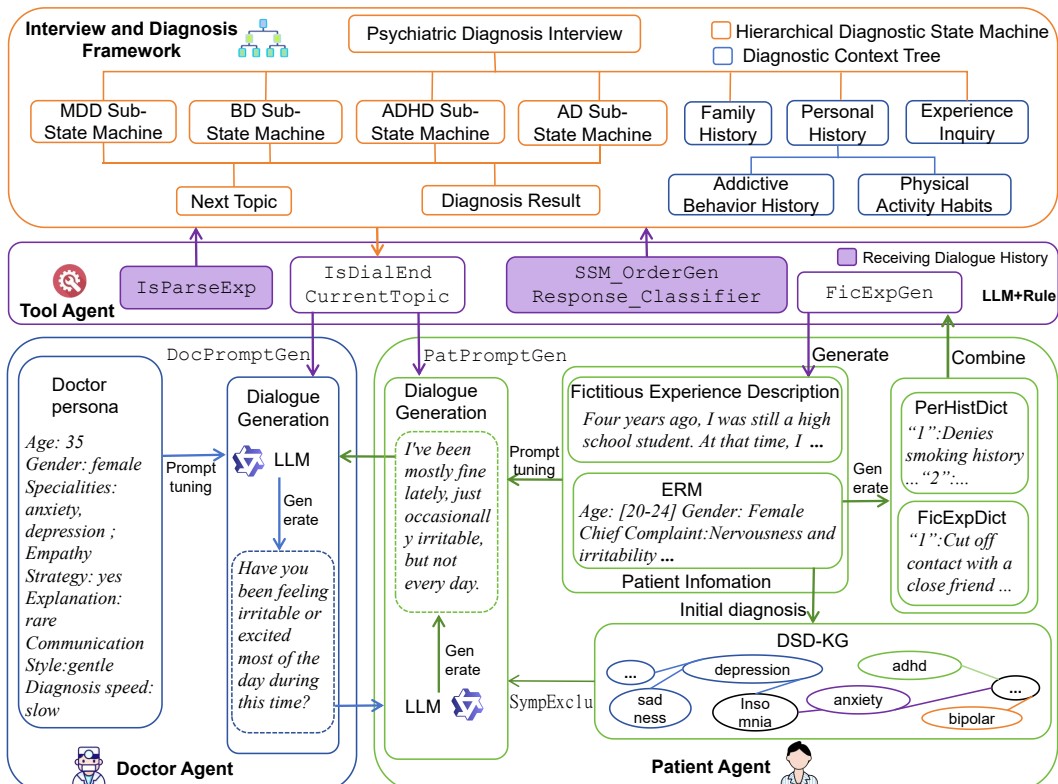

Figure 4: Multi-agent Diagnosis Framework. LLM-based doctor and patient agents interact under a tool agent that manages dialogue flow and diagnostic state transitions. The tool agent combines LLM and rule-based logic to generate patient experiences and regulate the Dynamic Diagnostic Tree.

- **High-Level States (HLS)**: Overarching modules (e.g., current-episode screening), shown as large *hollow rectangles* and represent overarching modules, such as current-episode screening (A.1) or episode history (A.15).

- **Intermediate-Level States (ILS)**: Group related symptoms under an HLS, displayed as *solid rectangles*. They group related symptoms under an HLS and typically have no fixed time window. A special subset of ILS, referred to as *sub-state groups*, is represented by *dashed rectangles*, and aggregates closely related questions for joint inquiry.

- **Basic-Level States (BLS)**: Terminal nodes corresponding to individual questions.drawn as *solid circles*. These are terminal nodes corresponding to individual questions, typically constrained by time and embedded within sub-state groups. Except for the dashed sub-state groups, each ILS and BLS node corresponds to a specific question derived from SCID-5-RV. Questions fall into one of

four categories: (i) affective or cognitive symptoms, (ii) physiological or behavioral changes, (iii) functional impairment or risk, and (iv) comorbid or contributing factors (for a full list of nodes and their categories, see Appendix A.5).

*Natural language cues.* Within each sub-state group, only the first question adopts a precise temporal phrase such as "in the past two weeks"; subsequent questions use looser expressions like "recently" to avoid unnatural repetition.

*Binary symptom scale and flow control.* The original four-point SCID-5-RV symptom scale ("unclear", "absent", "subthreshold", "threshold") is reduced to a binary form: *present* or *absent*. We do not discard severity information in this simplification; it is encoded in specific HDSM nodes and transitions (e.g., duration constraints such as A254 $\geq$ 1 week, and functional impairment checks such as A23 on school, work, or social functioning), while the binary variables only indicate whether diagnostic thresholds are met. We adopt this binarization not because the four-level severity scale is unimportant, but because distinctions such as "below threshold" versus "insufficient information" require nuanced clinical judgment: in our early experiments, asking LLM agents to maintain four-level severity ratings frequently produced inconsistent or contradictory outputs, leading to diagnostic loops or invalid transitions, whereas binary thresholding yielded a more stable and auditable diagnostic process. Colored arrows in Figure 5 illustrate how different responses guide the interview trajectory. If a BLS node has no valid successor, the agent performs a localized random traversal within the sub-state group. The transition to the next diagnostic state occurs only after all terminal BLS nodes (i.e., those without successors) have been visited. If the number of "positive" (i.e., "present") responses exceeds a threshold, the group is deemed *positive* and follows the corresponding path; otherwise, it is *absent* and triggers an alternative transition.

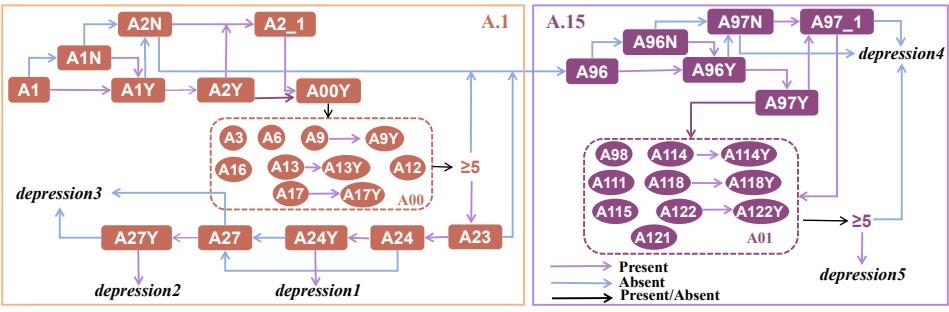

Figure 5: MDD sub-state machine. Solid nodes denote topic-specific questions. Colored arrows represent binary responses (*present* vs. *absent*), i.e., whether the patient exhibits the symptom. Groups A00 and A01 are activated when $\geq 5$ "present" responses are observed; otherwise, they follow the alternative transition.

**Diagnostic Context Tree (DCT)**    To further enhance the semantic depth and contextual coherence of diagnostic dialogues, we introduce the DCT, a tree-structured semantic controller that operates alongside the HDSM. The DCT is designed to dynamically manage the flow of the dialogue, ensuring that it remains clinically relevant and logically coherent. The top-level branches of the DCT include three main categories: *Family History*, *Personal History*, and *Experience Inquiry*. These categories are structured to capture the comprehensive background information necessary for a thorough psychiatric evaluation.

The *Experience Inquiry* node within the DCT is dynamically triggered at the end of each turn, based on the context of the conversation. This ensures that the dialogue remains responsive and adaptive to the specific needs of the interaction. After the completion of the HDSM, the remaining leaf nodes of the DCT are visited in a randomized order to maintain a natural flow of conversation. Personal history templates are conditionally selected based on patient gender, ensuring that the dialogue is tailored to the individual patient's background (see Figure 4 for male-specific examples).

By integrating the DCT with the HDSM, we effectively augment the elicitation of relevant background information, leading to dialogues that are more logically coherent and clinically aligned with real-world psychiatric consultations. This combination ensures that the dialogue not only fol-

lows a structured diagnostic process but also incorporates the necessary contextual details to support a comprehensive and accurate diagnosis.

## 4.2 MULTI-AGENT EXECUTION MECHANISM

Our simulator employs three specialized agents to facilitate each dialogue generation:

**Patient Agent**   The patient agent formulates responses based on (i) the structured EMR, (ii) its fictitious personal experience, and (iii) the current topic provided by the tool agent. Initial experiments revealed a bias towards affirming all symptoms, even those not documented in the EMR. To mitigate this, we developed a *Disease–Symptom Description Knowledge Graph* (DSD-KG) derived from SCID-5-RV guidelines. For each question, the agent consults the DSD-KG; if the symptom is absent from the EMR or contradicts the provisional diagnosis, the agent responds with *no*. This mechanism filters out hallucinated agreements, enhancing the credibility of the dialogue.

**Doctor Agent**   The doctor agent poses questions according to the topic queued by the tool agent. To avoid monotony, we defined five distinct doctor profiles varying in age, specialty, empathy style, verbosity, diagnostic speed, and explanation frequency. Profile cues and high-quality few-shot examples are injected at prompt time, enabling the LLM to adapt its responses (e.g., concise, gentle, or analytical). Reply length limits and a rotating pool of empathy phrases further reduce repetitive responses.

**Tool Agent**   The Tool Agent serves as the central controller, bridging natural-language dialogue with the symbolic diagnostic framework. Its responsibilities are divided into two main categories: tree management and dialogue coordination.

### Tree Management

• `Sub-State-Machine-OrderGen(conv)` → list: Determines the execution order of the four disorder-specific sub-state machines after the first dialogue turn. It supports two strategies: *random* shuffle or *symptom-informed* mode, which prioritizes modules based on current symptoms.

• `ResponseClassifier(conv)` → bool: Classifies the patient's latest answer as *present* or *absent* and forwards this label to the HDSM to trigger state transitions.

• `NeedExpBranch(conv)` → node: Decides whether to enter the "Experience Inquiry" branch of the context tree based on the previous turn's content.

### Dialogue Interface

• `BuildPrompt(topic)` → text: Generates aligned prompts for the doctor and patient agents based on the current topic node in the diagnostic tree, ensuring semantic consistency.

• `IsDialEnd(tree)` → bool: Determines whether the consultation should terminate. The dialogue ends when all disorder-specific sub-state machines reach their terminal states and all required nodes in the context tree—excluding the optional experience inquiry—have been traversed.

## 5   FINAL PRODUCT: PSYCOTALK DATASET

**Generation Process**   We begin with 502 structured EMRs from PsyCoProfile. For each EMR, we sample five distinct fictitious experiences from the generated $\mathcal{D}_{\text{fic}}$ and pair them with five different personal histories from $\mathcal{D}_{\text{his}}$, resulting in five *Fictitious Experience Descriptions* (FEDs) that capture patient-specific contextual narratives. Each FED is assigned a unique doctor profile that varies in empathy level, verbosity, and diagnostic pacing. All three agents—the doctor, patient, and tool agent—are instantiated using the Qwen2.5-72B model (Bai et al., 2023), deployed locally on a server with four NVIDIA A100-SXM4-80GB GPUs using `vLLM` (Kwon et al., 2023). For each FED and doctor profile pair, we generate two dialogues using different sub-state machine scheduling strategies: one that is symptom-informed and another that uses a randomly determined order. This process yields approximately 5,000 dialogues. From these, we select 3,000 dialogues whose final diagnostic labels match one of the six predefined comorbidity combinations in PsyCoProfile to form the final **PsyCoTalk** dataset. As all human evaluators are native Chinese speakers, we release only the Chinese version to ensure annotation quality. For comparative analysis, we include two addi-

tional Chinese diagnostic datasets, $D^4$ and **MDD-5k**, along with a de-identified clinical dialogue set (**Real-World Dial**) obtained from a psychiatric hospital. Dataset statistics are reported in Table 2.

Table 2: **Statistics of different datasets.** Avg. chars (D) and Avg. chars (P) measure the average Chinese characters per doctor's response and patient's response.

| Dataset | Avg. chars (D) | Avg. chars (P) | Avg. Turns | #Disorders | Comorbidity | #Dialogues |
|---|---|---|---|---|---|---|
| Real-World Dial | **28.3** | **35.8** | – | – | ✗ | – |
| $D^4$ | 20.4 | 14.9 | 21.6 | Depression | ✗ | 1,339 |
| MDD-5k | 91.1 | 162.8 | 26.8 | >25 | ✗ | **5,000** |
| PsyCoTalk | **34.0** | **43.5** | **45.9** | 4 | ✓ | 3,000 |

**Evaluation**   We conduct two types of human evaluation. First, 50 dialogues from PsyCoTalk are rated in a double-blind fashion by five licensed psychiatrists with more than seven years of clinical experience (randomly drawn from 3,000 dialogues, i.e., 1.67% of the corpus, with diagnostic and demographic ratios closely matching the full dataset: depression 0.35 vs. 0.38, anxiety 0.32 vs. 0.31, ADHD 0.21 vs. 0.19, bipolar 0.12 vs. 0.12, female 0.53 vs. 0.50; this evaluation scale is comparable to or more rigorous than prior clinician-based mental health LLM studies, which typically assess $\sim$0.15%–$\sim$2% of their corpora with 2–5 expert raters Qi et al. (2025); Yin et al. (2025); Gabriel et al. (2024); Yang et al. (2024)). Each dialogue is evaluated across six dimensions grouped under four criteria: *Professionalism*, *Communication*, *Fluency*, and *Realism*. Professionalism assesses whether the doctor successfully elicits all symptoms necessary for diagnosis. Communication evaluates (i) the doctor's proactivity in questioning and (ii) the patient's responsiveness. Fluency measures (i) syntactic and topical coherence and (ii) avoidance of redundancy. Realism assesses how closely the dialogue resembles actual psychiatric consultations. Second, we perform an AB test using 10 two-turn excerpts sampled from each dataset (PsyCoTalk, $D^4$, MDD-5k, and Real-World Dial). The same five psychiatrists are asked to judge whether each excerpt appears real or AI-generated. Each excerpt identified as "real" receives one point, and final realism scores are normalized to a 10-point scale.

**Objective Comparison**   As shown in Table 2, **PsyCoTalk** excels in scale and structural fidelity. It is the only dataset for psychiatric comorbidity, with 3,000 dialogues-over twice $D^4$ and close to MDD-5k. Each conversation averages **45.9** turns, nearly double other corpora. Utterance lengths are 34.0 characters (doctors) and 43.5 (patients), closest to real clinical conversations (28.3 and 35.8). By contrast, MDD-5k deviates most, with overly long utterances that reduce realism.

To verify the effectiveness of the fictitious personal experience module under EMR reuse and to validate the cross-lingual generalization of our pipeline, we assess dialogue diversity and conduct a small-scale English generation experiment compared with existing corpora; detailed metrics and results are reported in Appendix A.2.

To assess diagnostic accuracy, we use initial EMR labels as ground truth and evaluate with exact-match (all 5 disorders correct). A zero-shot Qwen2.5-72B baseline reaches 0.22 subset accuracy, while our HDSM-guided multi-agent system improves to **0.31**, a significant gain (McNemar's test, $p = 7 \times 10^{-6} < 0.001$). On 200 sampled cases, GPT-4o-mini and Deepseek-v3 achieve below 0.1, Qwen3-32B below 0.02, and Qwen3-8B below 0.04. Per-label F1: MDD (**0.92**), AD (**0.81**), ADHD (**0.64**), BD (**0.40**). These align with clinical trends where BD and ADHD are harder to diagnose than MDD and AD due to symptom overlap, subtler onset, and higher heterogeneity (Hui et al., 2018; Barkley & Brown, 2008).

Table 3: PsyCoTalk 6-dimension evaluation. (Prof. = professionalism; Comm.(i)/(ii) = communication; Flu.(i)/(ii) = fluency; Sim. = similarity)

| Prof. | Comm.(i) | Comm.(ii) | Flu.(i) | Flu.(ii) | Sim. |
|---|---|---|---|---|---|
| 7.72 | 8.14 | 8.24 | 7.42 | 6.79 | 6.67 |

Table 4: AB-test results. (Real = real-world dial)

| Dataset | Real | $D^4$ | MDD-5k | PsyCoTalk |
|---|---|---|---|---|
| **Score** | 6 | 4 | 1 | **5** |

**Subjective Results**   Table 3 summarizes expert evaluation results across six dimensions. Psy-CoTalk ranks highest in communication (8.14 for doctor initiative, 8.24 for patient engagement),

alongside strong scores in professionalism (7.72) and fluency (7.42 and 6.79), reflecting coherent and context-aware interactions. In the AB test for perceived realism, Table 4 shows that PsyCoTalk achieves a score of 5, second only to real-world data (6), indicating that its dialogue style closely approximates clinical expectations. By contrast, MDD-5k scores the lowest (1) due to its templated and repetitive utterances, which reduce perceived authenticity. These findings confirm that our HDSM-based multi-agent framework produces dialogues that are not only diagnostically informative but also linguistically coherent and clinically credible across diverse expert judgments.

## LIMITATIONS

PsyCoTalk focuses on four prevalent psychiatric disorders and their comorbidities, which reflects common real-world cases but limits coverage of rarer conditions. This choice is driven by the scarcity of clinically reliable comorbidity data, so we retain only combinations with sufficient symptom diversity and sample size to support stable EMR construction. While we conducted a small-scale English generation experiment, the main dataset remains Chinese due to expert-evaluation constraints, restricting multilingual applicability. Moreover, Reddit-derived symptom cues may introduce demographic or cultural skew, although the EMR template and SCID-5-based HDSM are language- and culture-agnostic and can be ported to local clinical corpora. Nevertheless, our pipeline is extensible and can be scaled to broader disorder coverage and cross-lingual settings. Future work will conduct more experiments and evaluations to quantify the dataset's impact on model performance and diagnostic reasoning.

## 6    CONCLUSION

In this paper, we introduce a two-stage pipeline that creates the first large-scale, clinically standard dataset for psychiatric comorbidity. First, we develop PsyCoProfile, converting social media posts into 502 structured electronic medical records that reflect real-world prevalence of six common disorder combinations. Second, we propose a multi-agent interview and diagnosis framework that transforms these records into PsyCoTalk, a corpus of 3,000 multi-turn dialogues mimicking clinical interviews. PsyCoTalk provides the necessary scale and detail to train models for multi-disorder screening.

## ETHICS STATEMENT

The authors have read and adhere to the ICLR Code of Ethics. This work does not involve human subjects, identifiable private data, or harmful applications. No real patient data are used; the only human involvement is expert evaluation by five licensed psychiatrists, who provided informed consent and were compensated according to responsible-research guidelines. No external sponsorship or conflict of interest influenced the design or conclusions of this work.

**Data provenance and privacy.** The raw posts that seed **PsyCoProfile** are drawn from the publicly available PsySym corpus, whose collection was approved by an institutional review board (IRB I2022158P) and complies with the Personal Information Protection Law of the People's Republic of China. All posts are anonymous, stripped of usernames, time stamps and locations, and stored under random identifiers. Our pipeline rewrites each post into a synthetic electronic medical record; no verbatim text that could re-identify an author is retained. The fictitious personal histories and life events are generated entirely by large language models. Five licensed psychiatrists reviewed random samples and confirmed that no record contains protected health information or disallowed content. The final dialogues in **PsyCoTalk** are synthetic and will be released only under a data-usage agreement that forbids attempts at re-identification or clinical deployment.

**Intended use and risk mitigation.** The dataset is designed for *research* on multi-disorder screening and dialogue modelling. It is *not* a diagnostic tool, nor does it provide treatment advice or actionable clinical guidance. To prevent misuse, we define compliant use boundaries as follows: (1) *Research-only restriction*: all EMRs and dialogues are synthetic and must not be used for individual clinical decisions, real-world psychological counseling, or any form of patient-facing service; (2) *Data-usage agreement*: access requires signing an agreement that prohibits clinical deployment, real-user psychological services, and any re-identification attempts. Any real-world mental-health use of

models trained on this dataset would require formal ethics review and clinician oversight. Synthetic conversations may still reflect biases inherited from web data and large language models; users must apply rigorous evaluation before any downstream use. We urge practitioners to keep humans in the loop when analysing sensitive mental-health text, and to follow relevant professional, legal and ethical guidelines. Future work will extend coverage while continuing to consult ethics boards and domain experts.

**Societal impact.** This work has the potential to positively contribute to early-stage screening research and the development of more inclusive diagnostic tools for psychiatric profiles, particularly in settings with limited clinical resources. However, the release of realistic synthetic dialogues and records also raises risks of misuse, such as inappropriate clinical adoption or amplification of model biases in downstream systems. To mitigate these risks, we emphasize that the dataset is strictly intended for research use and should be paired with responsible modeling and evaluation practices. Broader impacts will be monitored in future iterations in collaboration with ethics committees and clinical stakeholders.

**Annotator Compensation.** We ensured fair compensation for all annotators. They were compensated at a rate of $25/hour. This rate is above the local minimum wage and aligns with guidelines for ethical crowdsourcing practices. Annotators were briefed about the nature of the task, estimated time per task, and payment prior to participation.

## ACKNOWLEDGEMENT

This work has been supported by the China NSFC Project (No. 62572320), the Shanghai Shenkang Hospital Development Center Project (Grant No. SHDC12025118), Tianqiao and Chrissy Chen Institute (TCCI) and EverMind AI, Inc (Program: Mental Health Data Synthesis and Scientific Data Competition).

## REPRODUCIBILITY STATEMENT

We have made all code and datasets used in this work publicly available in an anonymous repository to ensure full reproducibility of our experiments: `https://github.com/X-LANCE/PsyCo-Diagnosis`.

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

# A  APPENDIX

## A.1  USE OF LLMS

LLMs were primarily employed for data generation, forming the basis of our multi-agent architecture. For the writing of this paper, we note that LLMs were used only to aid or polish the text, for example in grammar checking and improving readability. They were not involved in generating research ideas or analyzing results.

## A.2  DIALOGUE DIVERSITY AND CROSS-LINGUAL ANALYSIS

### A.2.1  INTRA-EMR DIVERSITY

To address concerns about EMR reuse, we enrich each dialogue with varied backgrounds while retaining symptom consistency. We measure intra-EMR diversity by $1 - \mathrm{J}(EMR_i)$, where

$$\mathrm{J}(EMR_i) = \frac{2}{n(n-1)} \sum_{j<k} \frac{|K_j \cap K_k|}{|K_j \cup K_k|},$$

with $K_j, K_k$ denoting keyword sets from dialogues generated under the same EMR $i$. The average diversity score is $0.647 \pm 0.030$, i.e., only 35.3% overlap, demonstrating high variability among dialogues derived from the same case.

A.2.2 COMPARISON WITH REAL CLINICAL DIALOGUES

We further compare synthetic (PsyCoTalk) and real-world clinical dialogues (1,731 patients, with 40.7% comorbid cases) using three diversity metrics:

1. **Normalized Entropy** $H(X)/\log_2(V)$, where $H(X)$ is Shannon entropy and $V$ is vocabulary size.

2. **Hapax Proportion** $N_{\text{hapax}}/V$, where $N_{\text{hapax}} = |\{w_i \in V \mid f(w_i) = 1\}|$.

3. **Semantic Diversity** $1 - \text{MeanCosSim}$, with $\text{MeanCosSim} = \frac{2}{n(n-1)} \sum_{i<j} \cos(\mathbf{v}_i, \mathbf{v}_j)$, where each session $s_i$ is represented by an embedding vector $\mathbf{v}_i$.

Table 5 shows that PsyCoTalk (Chinese) closely matches the diversity of real clinical dialogues, while PsyCoTalk-eng exhibits comparable or even higher diversity than the AlexanderStreet dataset.

Table 5: Diversity metrics across synthetic and real dialogue datasets.

| Metric | PsyCoTalk (Chinese, 3k) | Real Clinical (Chinese, 1.7k) | PsyCoTalk-eng (205) | AlexanderStreet (English, 1.3k) |
|---|---|---|---|---|
| Normalized Entropy | 0.5974 | 0.5880 | **0.6938** | 0.5722 |
| Hapax Proportion | 0.3231 | **0.4195** | 0.2189 | 0.3143 |
| Semantic Diversity | 0.7938 | 0.8663 | **0.9746** | 0.9678 |

A.2.3 ENGLISH DATA AND CROSS-LINGUAL COMPARISON

Using the same pipeline, we construct a preliminary English dataset (PsyCoTalk-eng, 205 dialogues from 41 patients) and compare it with the AlexanderStreet dataset (Alex, 1,254 dialogues without diagnostic labels). As shown in Table 5, PsyCoTalk-eng achieves higher normalized entropy and semantic diversity than Alex, while Chinese PsyCoTalk exhibits a higher Hapax proportion but lower semantic diversity than its English counterpart. This discrepancy may be attributed to language structure: Chinese tends to use more unique or low-frequency words, while English dialogues often display greater semantic variation.

A.2.4 WORD FREQUENCY AND PART-OF-SPEECH COMPARISON

We compare word frequency distributions in 200 sampled English and 200 Chinese dialogues, tokenized by NLTK (English) and jieba (Chinese). The top 100 high-frequency words (frequency $> 100$) are categorized into four groups, and the results are summarized in Table 6. Here, *High-freq Count (%)* refers to the number and proportion of distinct word types from each category that appear among the top 100 (e.g., 28 Chinese words fall into "Symptoms/States," accounting for 35%). *Total Freq (%)* denotes the cumulative frequency and proportion of all words in that category within the sampled dialogues (e.g., Chinese symptom/state words occur 27,342 times in total, 21.88% of all top-100 occurrences). This dual perspective captures both the lexical coverage (Count) and the usage weight (Total Freq) of different categories across languages.

Table 6: Word frequency comparison across Chinese and English dialogues.

| Word Class | Chinese High-freq Count (%) | English High-freq Count (%) | Chinese Total Freq (%) | English Total Freq (%) |
|---|---|---|---|---|
| Symptoms/States | 28 (35%) | 26 (26%) | 27,342 (21.88%) | 39,553 (21.45%) |
| Descriptive Words | 39 (48.8%) | 62 (62%) | 49,305 (39.45%) | 113,670 (61.64%) |
| Time | 19 (23.8%) | 9 (9%) | 30,702 (24.57%) | 20,895 (11.33%) |
| Degree | 14 (17.5%) | 3 (3%) | 17,632 (14.11%) | 10,310 (5.59%) |

Although distributional differences exist, both languages are dominated by descriptive and symptom-related words, with similar category rankings. Figure 6 illustrates representative word

clouds for the "symptom/state" category in both languages, showing numerous synonymous or semantically close pairs (e.g., *anxious*/烦躁, *mind*/心里/脑子里). English words exhibit a more even distribution, potentially due to linguistic structural differences.

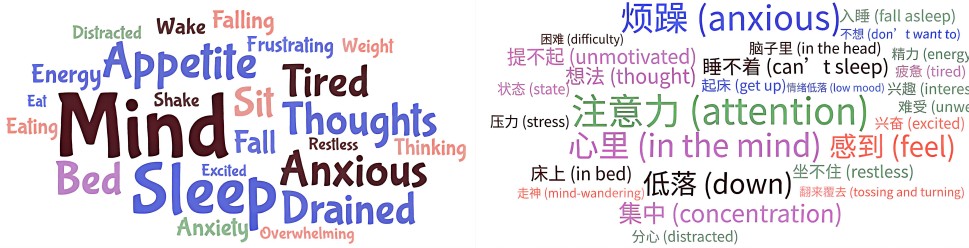

Figure 6: Word clouds of top "symptom/state" high-frequency words in English (left) and Chinese (right).

## A.3 EMR SECTION GUIDELINES

This appendix provides detailed specifications for each section in the standardized EMR template developed in this study. The content structure was developed based on a standard clinical case template obtained directly from practicing psychiatrists. This template reflects real-world documentation practices used in psychiatric settings and was adopted to ensure clinical realism and consistency.

***Demographic Information*** should include the patient's gender, age, educational background, marital status, and occupation. The language should be concise and factual.

***Chief Complaint*** describes the primary psychological symptoms reported by the patient, along with their duration. The text should be brief, focused, and clinically relevant.

***Medical Condition*** covers the current psychiatric presentation in more detail. This includes the onset and course of symptoms, any identifiable triggers, and the impact on the patient's daily functioning. General physical health indicators such as sleep patterns, appetite, and weight changes should also be summarized when available.

***Medical History*** lists prior medical conditions including chronic diseases, previous hospitalizations, surgical procedures, known allergies, and history of infections. This section helps assess comorbid risks.

***Personal History*** records health-related lifestyle factors such as smoking, alcohol use, and drug consumption. In female patients, it may also include relevant information on menstruation.

***Family History*** identifies psychiatric or neurological disorders in close biological relatives (e.g., parents, siblings). Information should be structured to reflect genetic predisposition or family dynamics relevant to diagnosis.

***Preliminary Diagnosis*** provides an initial clinical impression based on the information presented in the EMR. This diagnosis is tentative and may refer to DSM-based categories such as MDD, BD, or ADHD, depending on symptom patterns.

## A.4 PROMPT DETAILS FOR VIRTUAL EXPERIENCE GENERATION

To enrich patient narratives with personalized and semantically coherent context, we design a structured prompt-based mechanism for generating *Fictitious Experience Descriptions* . The generation is executed by the Tool Agent using an instruction-following large language model.

The input to this process is a template sentence derived from each patient's structured EMR, formatted as: *"I am a {age}-year-old {gender} with {diagnosis}, currently working or studying as a {occupation}. Past experience: {experience}"*. The fields *{age}*, *{gender}*, *{diagnosis}*, and *{occupation}*

are directly drawn from the EMR, while *{experience}* is sampled from the fictitious experience dictionary $\mathcal{D}_{\text{fic}}$.

The prompt guides the model to craft a first-person narrative that elaborates on the given background, using the following system message:

> "You are a powerful, imaginative text assistant skilled at writing stories. Given keywords describing a patient's mental health background and experiences, imagine and write a detailed, first-person account of the patient's past. Expand on the causes, progression, and consequences of a psychologically impactful event. Do not use metaphors, exaggerated language, or fictitious names. Avoid therapeutic expressions like 'under the doctor's guidance' or 'trying to get better.' The output should be a coherent and realistic narrative."

An example output is shown below :

> *Four years ago, I was a 12-year-old middle school student. I was cheerful by nature, but often had trouble focusing. During class, I couldn't help but look around, and my hands were constantly doodling on paper. The teacher was very displeased and reminded me multiple times to pay attention, but it barely worked. One day in math class, the teacher was explaining a complicated geometry problem. I had stayed up late the night before playing video games, and couldn't follow the lesson at all. My attention drifted, and the teacher's voice gradually became a blur. Suddenly, the teacher raised his voice, "You! Stand up!" I was startled, jumped to my feet, heart pounding, and my face flushed with embarrassment. He stared at me sternly and said in front of the entire class, "You're not listening again, are you? With that kind of attitude, how do you expect to succeed? Do you think your parents would be proud of this?" The room fell silent. All eyes were on me. I felt utterly humiliated, and tears welled up in my eyes. The teacher went on: "If you don't take this seriously, you'll never amount to anything." His words pierced through me like needles. I felt defeated and deeply ashamed. After class, students whispered among themselves—some mocked me, others looked at me with pity. I locked myself in the bathroom and cried uncontrollably. For a while after that, I began to fear going to school. Just entering the classroom would make my palms sweat, and I couldn't concentrate at all. My grades plummeted. My parents, anxious and disappointed, constantly scolded me for not studying properly. A few months later, I was diagnosed with depression and ADHD. The doctor said that the public criticism had dealt a severe psychological blow, and combined with academic pressure and high family expectations, it ultimately triggered the onset of my condition.*

This prompt design ensures that generated *Fictitious Experience Descriptions* are aligned with EMR content, emotionally rich, and free from contradictions. It balances creative storytelling with medical plausibility, enabling the system to simulate realistic psychological histories without relying on deterministic templates.

## A.5 HIERARCHICAL DIAGNOSTIC STATE TAXONOMY AND TOPIC MAPPING

This appendix collates the full hierarchy of the HDSM, aligning each Basic-, Intermediate-, and High-Level State with its corresponding clinical topic. States are grouped according to the four SCID-5–derived categories:

(i) Affective & Cognitive Symptoms, (ii) Physiological & Behavioral Changes, (iii) Functional Impairment & Risk, and (iv) Comorbid or Contributing Factors. For completeness, terminal nodes that represent each sub-state machine's final diagnostic outcomes are also listed. The lists that follow serve as a reference for all experiments, ensuring that every state's semantic scope—and its precise role in the dialogue-generation pipeline—is clearly documented.

**I. MDD Sub-State Machine (including HBL A.1, A.15)**

A.1: Current Major-Depressive-Episode Screening

| Full Topic Inventory of the HBL A.1 Categorized by Four Clinical Dimensions |
| --- |
| **(i) Affective & Cognitive Symptoms**
• A1: Depressed mood • A1N: Sadness / emptiness / hopelessness
• A1Y: Duration $\geq$ 2 weeks • A2Y: Recent loss of interest / pleasure
• A2N: Reduced interest / pleasure • A2_1: Duration $\geq$ 2 weeks |
| **(ii) Physiological & Behavioral Changes**
• A3: Appetite / weight change • A6: Sleep disturbance
• A9: Psychomotor retardation / agitation • A9Y: Observable severity
• A12: Fatigue / loss of energy • A13: Worthlessness / excessive guilt
• A13Y: Functional limitation • A16: Poor concentration / indecisiveness |
| **(iii) Functional Impairment & Risk**
• A17: Suicidal ideation • A17Y: Suicide plan or behavior
• A23: Functional impairment |
| **(iv) Comorbid or Contributing Factors**
• A24: History of medical illness • A24Y: Related to medical illness
• A27: History of substance use • A27Y: Substance-related |

A.15: History of Major-Depressive Episodes

| Full Topic Inventory of the HBL A.15 Categorized by Three Clinical Dimensions |
| --- |
| **(i) Affective & Cognitive Symptoms**
• A96: Past depressed mood • A96N: Past sadness / emptiness / hopelessness
• A96Y: Duration $\geq$ 2 weeks • A97Y: Time-specific loss of interest / pleasure
• A97N: Past loss of interest / pleasure • A97_1: Duration $\geq$ 2 weeks |
| **(ii) Physiological & Behavioral Changes**
• A98: Appetite / weight change • A111: Sleep disturbance
• A114: Psychomotor retardation or agitation • A114Y: Observable severity
• A117: Fatigue / loss of energy • A118: Worthlessness / excessive guilt
• A118Y: Functional limitation • A121: Poor concentration / indecisiveness |
| **(iii) Functional Impairment & Risk**
• A122: Suicidal ideation • A122Y: Suicide plan or behavior |

| Final-State | Diagnosis Description |
| --- | --- |
| `depression1` | Major depressive episode due to physical illness |
| `depression2` | Major depressive episode induced by substances or medication |
| `depression3` | Primary major depressive episode |
| `depression4` | No major depressive disorder |
| `depression5` | Past major depressive episode |

**II. BD Sub-State Machine (including HBL A.23, A.43, D.1)**

A.23: Current Manic/Hypomanic Episode

---

**Full Topic Inventory of the HBL A.23 Categorized by Four Clinical Dimensions**

**(i) Affective & Cognitive Symptoms**
- A134: Elevated / manic mood    • A136: Irritability
- A135: Beyond normal range    • A137: Duration ≥ 1 week
- A137N: Irritability ≥ 1 week    • A137Y: Symptoms nearly every day
- A02Y (Sub-state A02): Worst week of episode    • A138 (Sub-state A02): Grandiosity / inflated self-esteem
- A141 (Sub-state A02): Flight of ideas    • A142 (Sub-state A02): Distractibility

**(ii) Physiological & Behavioral Changes**
- A135_1: Excessive energy / increased activity    • A136Y: Euphoria / high energy
- A139 (Sub-state A02): Decreased need for sleep    • A140 (Sub-state A02): Talkativeness
- A143 (Sub-state A02): Increased work drive / involvement    • A144 (Sub-state A02): Increased social activity
- A145 (Sub-state A02): Increased sexual activity    • A146 (Sub-state A02): Impulsive behavior
- A147 (Sub-state A02): Restlessness

**(iii) Functional Impairment & Risk**
- A148: Functional impairment

**(iv) Comorbid or Contributing Factors**
- A151: History of medical illness    • A151Y: Related to medical illness
- A154: History of substance use    • A154Y: Substance-related

---

A.43: History of Manic/Hypomanic Episodes

---

**Full Topic Inventory of the HBL A.43 Categorized by Two Clinical Dimensions**

**(i) Affective & Cognitive Symptoms**
- A251: Past elevated / manic mood    • A253: Past irritability
- A252: Beyond normal range    • A254: Duration ≥ 1 week
- A253N: Irritability ≥ 1 week    • A254Y: Symptoms nearly every day
- A255 (Sub-state A03): Grandiosity / inflated self-esteem    • A258 (Sub-state A03): Flight of ideas
- A259 (Sub-state A03): Distractibility    • A286 (Sub-state A04): Grandiosity / inflated self-esteem
- A289 (Sub-state A04): Flight of ideas    • A290 (Sub-state A04): Distractibility

**(ii) Physiological & Behavioral Changes**
- A252_1: Excessive energy / increased activity    • A253Y: Euphoria / high energy
- A256 (Sub-state A03): Decreased need for sleep    • A257 (Sub-state A03): Talkativeness
- A260 (Sub-state A03): Increased work drive / involvement    • A261 (Sub-state A03): Increased social activity
- A262 (Sub-state A03): Increased sexual activity    • A263 (Sub-state A03): Impulsive behavior
- A264 (Sub-state A03): Restlessness    • A287 (Sub-state A04): Decreased need for sleep
- A288 (Sub-state A04): Talkativeness    • A291 (Sub-state A04): Increased work drive / involvement
- A292 (Sub-state A04): Increased social activity    • A293 (Sub-state A04): Increased sexual activity
- A294 (Sub-state A04): Impulsive behavior    • A295 (Sub-state A04): Restlessness

---

D.1: Determinative Clauses for Bipolar Diagnosis

| Label | Description |
|-------|-------------|
| D3 | At least one manic episode |
| D5 | At least one hypomanic and one major-depressive episode |

| Final-State | Diagnosis Description |
|-------------|----------------------|
| bipolar1 | Hypomanic episode |
| bipolar2 | Manic episode due to physical illness |
| bipolar3 | Manic episode induced by substances or medication |
| bipolar4 | Primary manic episode |
| bipolar5 | No manic or hypomanic disorder |
| bipolar6 | Past manic episode |
| bipolar7 | Past hypomanic episode |
| bipolar8 | Bipolar I disorder |
| bipolar9 | Bipolar II disorder |

## III. AD State Machine (including HBL F.25, F.31)

F.25: Current Generalized-Anxiety-Disorder Symptoms

---

**Full Topic Inventory of the HBL F.25 Categorized by Four Clinical Dimensions**

**(i) Affective & Cognitive Symptoms**
• F140: Persistent anxiety / worry    • F142_1: Unfounded anxiety / excessive worry
• F142_2: Anxiety present most of $\geq$ 6 months    • F143: Difficulty controlling worry

**(ii) Physiological & Behavioral Changes**
• F144 (Sub-state F00): Restlessness / on edge    • F145 (Sub-state F00): Easy fatigability
• F146 (Sub-state F00): Difficulty concentrating / mind going blank    • F147 (Sub-state F00): Irritability
• F148 (Sub-state F00): Muscle tension    • F149 (Sub-state F00): Sleep disturbance / tiredness

**(iii) Functional Impairment & Risk**
• F151: Functional impairment (study, work, social)    • F163: Panic attacks

**(iv) Comorbid or Contributing Factors**
• F152: History of medical illness    • F152Y: Related to medical illness
• F156: History of substance/medication use    • F156Y: Substance-related

---

F.31: History of Generalized-Anxiety Disorder

---

**Full Topic Inventory of the HBL F.31 Categorized by Two Clinical Dimensions**

**(i) Affective & Cognitive Symptoms**
• F165: Months-long anxiety / worry    • F167_1: Unfounded anxiety / excessive worry
• F167_2: Duration $\geq$ 6 months    • F168: Difficulty controlling worry

**(ii) Physiological & Behavioral Changes**
• F169 (Sub-state F01): Restlessness / on edge    • F170 (Sub-state F01): Easy fatigability
• F171 (Sub-state F01): Difficulty concentrating / mind going blank    • F172 (Sub-state F01): Irritability

---

*Continued on next page*

*Continued from previous page*

| | |
|---|---|
| • F173 (Sub-state F01): Muscle tension | • F174 (Sub-state F01): Sleep disturbance / tiredness |

| Final-State | Diagnosis Description |
|---|---|
| `anxiety1` | Generalized anxiety disorder (GAD) due to physical illness |
| `anxiety2` | GAD induced by substances or medication |
| `anxiety3` | Current GAD with panic attacks |
| `anxiety4` | Current generalized anxiety disorder |
| `anxiety5` | No generalized anxiety disorder |
| `anxiety6` | Past generalized anxiety disorder |

## IV. ADHD State Machine (including HBL K.1)

K.1: Comprehensive ADHD Symptom Assessment

**Full Topic Inventory of the HBL K.1 Categorized by Three Clinical Dimensions**

**(i) Affective & Cognitive Symptoms**

• K2: Easily distracted / disorganized          • K6 (Sub-state K00): Frequently misses details / careless work

• K6N (Sub-state K00): Careless mistakes (e.g., billing)          • K7 (Sub-state K00): Difficulty sustaining attention (reading, conversations, chores)

• K8 (Sub-state K00): Seems not to listen when spoken to          • K8Y (Sub-state K00): Frequency of inattention

• K9 (Sub-state K00): Leaves tasks unfinished due to distraction          • K12 (Sub-state K00): Often loses/misplaces items

• K13 (Sub-state K00): Easily distracted by external stimuli          • K13N (Sub-state K00): Distracted by unrelated thoughts

• K14 (Sub-state K00): Forgetful (e.g., returning calls, paying bills)          • K11 (Sub-state K00): Avoids tasks requiring sustained attention

**(ii) Physiological & Behavioral Changes**

• K2N: Difficulty sitting still / waiting in lines          • K16 (Sub-state K01): Restless when required to sit

• K17 (Sub-state K01): Leaves seat (class, cinema) frequently          • K18 (Sub-state K01): Feeling restless when inactive

• K19 (Sub-state K01): Cannot engage quietly in leisure          • K19N (Sub-state K01): Talkative/noisy when should be quiet

• K20 (Sub-state K01): "On the go"; exhausting for others          • K21 (Sub-state K01): Talks excessively / complaints of verbosity

• K22 (Sub-state K01): Often blurts out answers          • K23 (Sub-state K01): Difficulty waiting one's turn

• K24 (Sub-state K01): Interrupts or intrudes on others

**(iii) Functional Impairment & Risk**

• K10 (Sub-state K00): Difficulty organizing tasks at home or work          • K10N (Sub-state K00): Extremely messy desk or closet

• K5: Symptoms persisting in the past 6 months

| Final-State | Diagnosis Description |
|---|---|
| adhd1 | No attention-deficit/hyperactivity disorder (ADHD) |
| adhd2 | Attention-deficit/hyperactivity disorder |

## A.6 CASE STUDY OF PSYCOPROFILE

To aid interpretability, we provide an English translation of a structured EMR originally generated in Chinese. As shown in Figure 7, the record describes a 20–24-year-old unmarried female student with a history of major depressive disorder and generalized anxiety.

The chief complaint includes emotional instability, fatigue, and occasional negative thoughts. Her medical condition outlines symptom progression over 28 months, including episodes of social withdrawal, panic attacks, sleep disturbances, and compulsive behavior, with gradual improvement following psychological intervention. Her past medical history includes SSRI treatment, hospitalization, and CBT, as well as group-based mindfulness therapy and weekly psychiatric supervision.

Her family history is positive for bipolar disorder (mother) and antisocial personality disorder (father). Personal history reports menstrual irregularities, no smoking, regular caffeine intake, and a preference for outdoor exercise. Demographics indicate a university education and no history of marriage. The preliminary diagnosis is depression and anxiety.

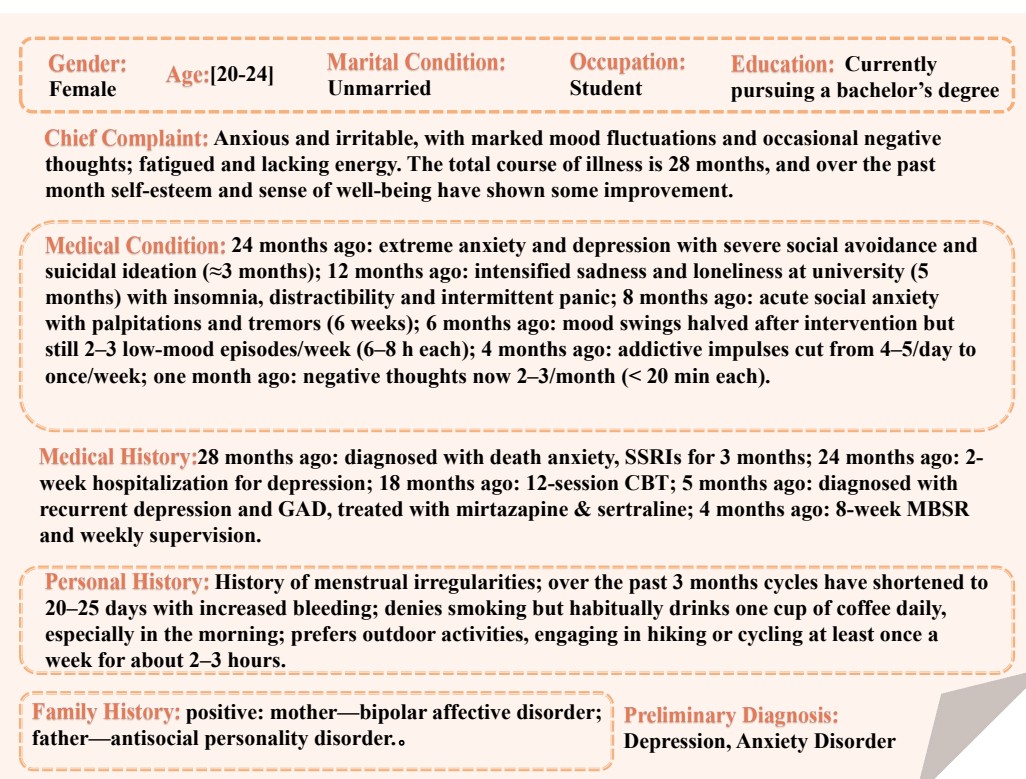

Figure 7: Example of a Structured EMR (English Version))

## A.7 HDSM FRAMEWORK

Figure 5 presents the full structure of the Hierarchical Diagnostic State Machine (HDSM), covering all four targeted psychiatric disorders. Each disorder is assigned an independent sub-state machine, with corresponding high-level states (HLS) shown in the figure: A.1 and A.15 for MDD, F.25 and F.31 for AD, A.1, A.43, and D.1 for BD, and K.1 for ADHD.

Each HLS contains one or two sub-state groups (dashed rectangles), except D.1, which serves as a summary node in the BD sub-state machine. Sub-state groups are designed based on SCID-5-RV item clusters, preserving clinical logic and content alignment. Each sub-state group has a predefined threshold, typically 3 or 5, for determining whether the collected responses support a *positive* diagnostic transition. This design enables the system to emulate real-world psychiatric evaluations with high clinical fidelity.

Terminal states, such as depression1 through depression5 in the MDD branch, represent mutually exclusive diagnostic outcomes based on structured symptom elicitation. All state transitions, node categories, and thresholds strictly follow the SCID-5-RV guidelines, ensuring consistency between the simulated dialogue flow and professional diagnostic protocols.

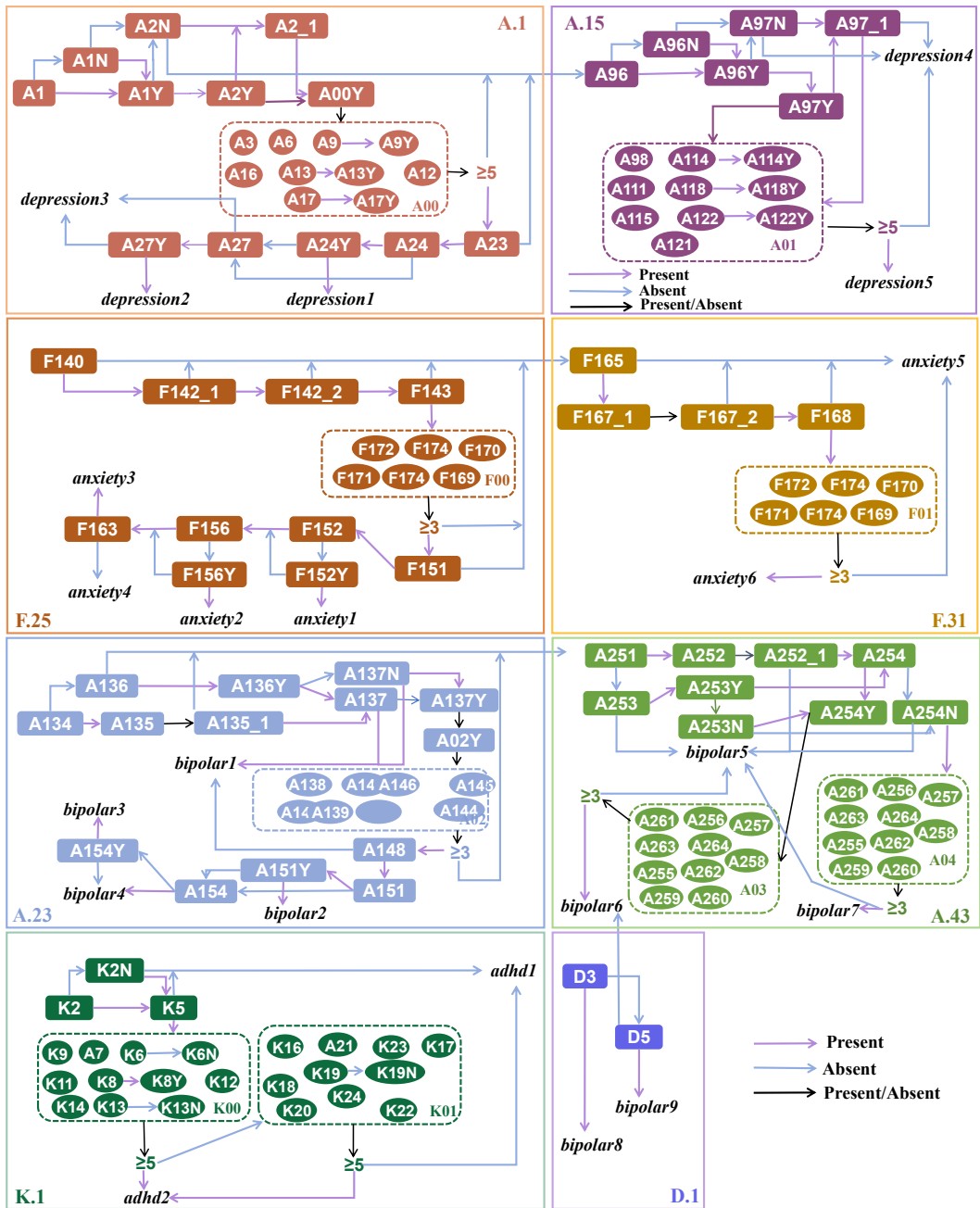

Figure 8: Overview of the HDSM. High-level states A.1 and A.15 correspond to the MDD sub-state machine; F.25 and F.31 to the AD sub-state machine; A.1, A.43, and D.1 to the BD sub-state machine; and K.1 to the ADHD sub-state machine.

## A.8  CASE STUDY OF PSYCOTALK

To illustrate the structure and quality of PsyCoTalk, we present an example doctor–patient dialogue in both Chinese (original) and English (translated) versions. These dialogues are generated by multi-agent interaction under the Diagnosis-Agents framework, where the patient agent is instantiated with a personalized EMR and fictitious experience, and the doctor agent is guided by a structured diagnostic state machine.

The dialogue includes a complete diagnostic process, from initial inquiry to the final diagnosis issued by the doctor agent. We present below an English translation of the original dialogues for illustrative purposes (see Figure 9–Figure 14).

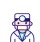 **Doctor:** How have you been feeling lately? Is there anything in particular that's been bothering you?

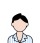 **Patient:** I've been sleeping okay, but when I wake up I feel completely drained, as if I haven't slept enough⋯Studying feels tough; after reading for a bit my mind just stops working, and I often drift off thinking about random things⋯I don't even know what I'm thinking about⋯

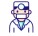 **Doctor:** Besides zoning out during study, are you able to focus on other daily activities?

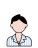 **Patient:** I get distracted with other things too⋯Yesterday while running, my mind kept racing about homework and I couldn't keep going⋯It feels like I can't concentrate on anything for very long⋯

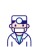 **Doctor:** How long has this difficulty focusing lasted? Have you noticed any significant mood swings?

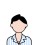 **Patient:** It's been about three months, and my mood's been pretty unstable. Sometimes I suddenly feel very down, like I can't muster the energy to do anything—even my favorite games don't appeal to me.

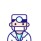 **Doctor:** Did anything significant happen three months ago that caused you a lot of stress? When you feel low, how long does it usually last?

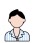 **Patient:** That was finals time plus a lot going on at home, so the stress was intense. When I felt down, sometimes I couldn't shake it for a whole day—food tasted bland.

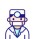 **Doctor:** Is your stress still high these days? Besides loss of appetite, any other physical discomfort?

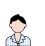 **Patient:** It's a bit better now, but I still feel dizzy and unsteady when walking. Last night I couldn't sleep until 2 AM.

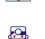 **Doctor:** Is insomnia at night frequent? When did the dizziness start?

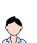 **Patient:** The dizziness began around last weekend, and tossing and turning at night has been common.

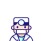 **Doctor:** Have you experienced a similar situation since that last finals period in the past six months?

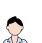 **Patient:** Not as severe as then, but I still get anxious under study stress. Coffee doesn't help—I feel foggy and my efficiency is low.

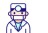 **Doctor:** Have you felt tense and uneasy every day recently?

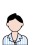 **Patient:** Yes, my palms sweat in class, I lack appetite, and I often wake up at night and can't get back to sleep.

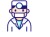 **Doctor:** After waking up, what thoughts run through your mind?

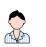 **Patient:** I think of past exam failures, feel I didn't study enough, and only calm down after checking my books.

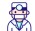 **Doctor:** Does that checking calm you down?

Figure 9: Full doctor–patient dialogue in English (Page 1)

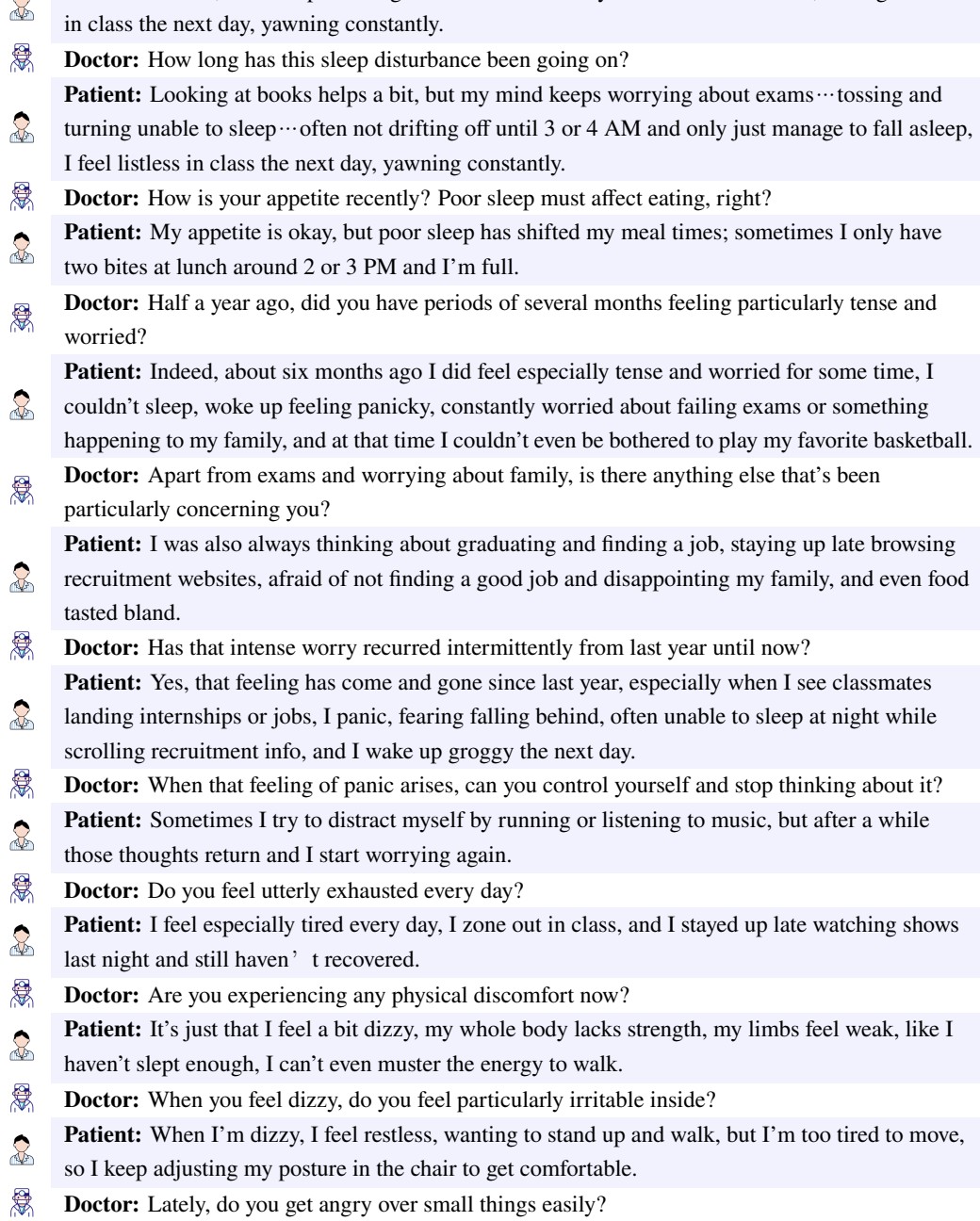

**Patient:** A little, but I keep thinking about exams and only drift off around dawn, feeling listless in class the next day, yawning constantly.

**Doctor:** How long has this sleep disturbance been going on?

**Patient:** Looking at books helps a bit, but my mind keeps worrying about exams···tossing and turning unable to sleep···often not drifting off until 3 or 4 AM and only just manage to fall asleep, I feel listless in class the next day, yawning constantly.

**Doctor:** How is your appetite recently? Poor sleep must affect eating, right?

**Patient:** My appetite is okay, but poor sleep has shifted my meal times; sometimes I only have two bites at lunch around 2 or 3 PM and I'm full.

**Doctor:** Half a year ago, did you have periods of several months feeling particularly tense and worried?

**Patient:** Indeed, about six months ago I did feel especially tense and worried for some time, I couldn't sleep, woke up feeling panicky, constantly worried about failing exams or something happening to my family, and at that time I couldn't even be bothered to play my favorite basketball.

**Doctor:** Apart from exams and worrying about family, is there anything else that's been particularly concerning you?

**Patient:** I was also always thinking about graduating and finding a job, staying up late browsing recruitment websites, afraid of not finding a good job and disappointing my family, and even food tasted bland.

**Doctor:** Has that intense worry recurred intermittently from last year until now?

**Patient:** Yes, that feeling has come and gone since last year, especially when I see classmates landing internships or jobs, I panic, fearing falling behind, often unable to sleep at night while scrolling recruitment info, and I wake up groggy the next day.

**Doctor:** When that feeling of panic arises, can you control yourself and stop thinking about it?

**Patient:** Sometimes I try to distract myself by running or listening to music, but after a while those thoughts return and I start worrying again.

**Doctor:** Do you feel utterly exhausted every day?

**Patient:** I feel especially tired every day, I zone out in class, and I stayed up late watching shows last night and still haven't recovered.

**Doctor:** Are you experiencing any physical discomfort now?

**Patient:** It's just that I feel a bit dizzy, my whole body lacks strength, my limbs feel weak, like I haven't slept enough, I can't even muster the energy to walk.

**Doctor:** When you feel dizzy, do you feel particularly irritable inside?

**Patient:** When I'm dizzy, I feel restless, wanting to stand up and walk, but I'm too tired to move, so I keep adjusting my posture in the chair to get comfortable.

**Doctor:** Lately, do you get angry over small things easily?

Figure 10: Full doctor–patient dialogue in English (Page 2)

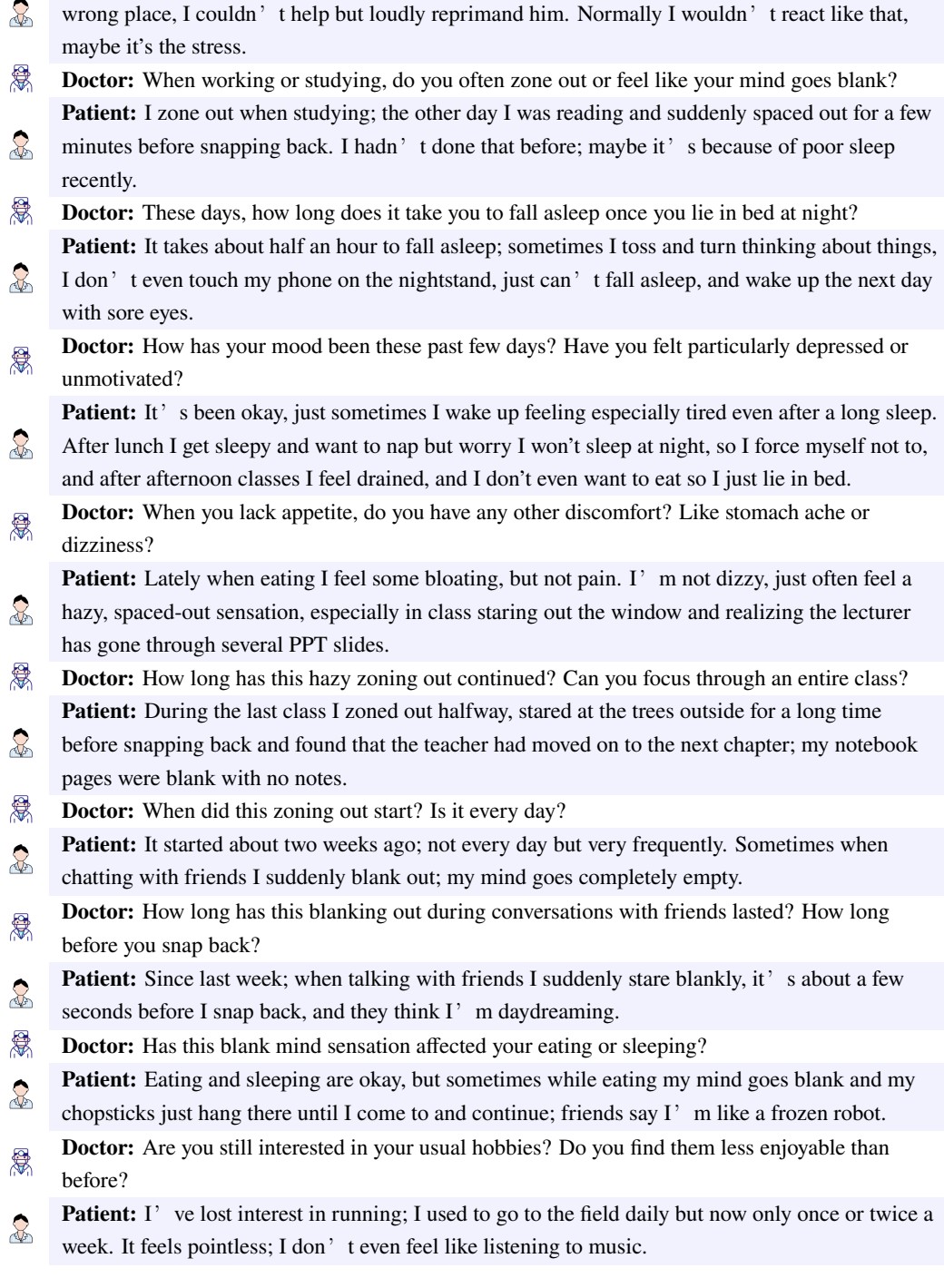

**Patient:** Recently I do get a bit irritable; last week, because my roommate threw garbage in the wrong place, I couldn't help but loudly reprimand him. Normally I wouldn't react like that, maybe it's the stress.

**Doctor:** When working or studying, do you often zone out or feel like your mind goes blank?

**Patient:** I zone out when studying; the other day I was reading and suddenly spaced out for a few minutes before snapping back. I hadn't done that before; maybe it's because of poor sleep recently.

**Doctor:** These days, how long does it take you to fall asleep once you lie in bed at night?

**Patient:** It takes about half an hour to fall asleep; sometimes I toss and turn thinking about things, I don't even touch my phone on the nightstand, just can't fall asleep, and wake up the next day with sore eyes.

**Doctor:** How has your mood been these past few days? Have you felt particularly depressed or unmotivated?

**Patient:** It's been okay, just sometimes I wake up feeling especially tired even after a long sleep. After lunch I get sleepy and want to nap but worry I won't sleep at night, so I force myself not to, and after afternoon classes I feel drained, and I don't even want to eat so I just lie in bed.

**Doctor:** When you lack appetite, do you have any other discomfort? Like stomach ache or dizziness?

**Patient:** Lately when eating I feel some bloating, but not pain. I'm not dizzy, just often feel a hazy, spaced-out sensation, especially in class staring out the window and realizing the lecturer has gone through several PPT slides.

**Doctor:** How long has this hazy zoning out continued? Can you focus through an entire class?

**Patient:** During the last class I zoned out halfway, stared at the trees outside for a long time before snapping back and found that the teacher had moved on to the next chapter; my notebook pages were blank with no notes.

**Doctor:** When did this zoning out start? Is it every day?

**Patient:** It started about two weeks ago; not every day but very frequently. Sometimes when chatting with friends I suddenly blank out; my mind goes completely empty.

**Doctor:** How long has this blanking out during conversations with friends lasted? How long before you snap back?

**Patient:** Since last week; when talking with friends I suddenly stare blankly, it's about a few seconds before I snap back, and they think I'm daydreaming.

**Doctor:** Has this blank mind sensation affected your eating or sleeping?

**Patient:** Eating and sleeping are okay, but sometimes while eating my mind goes blank and my chopsticks just hang there until I come to and continue; friends say I'm like a frozen robot.

**Doctor:** Are you still interested in your usual hobbies? Do you find them less enjoyable than before?

**Patient:** I've lost interest in running; I used to go to the field daily but now only once or twice a week. It feels pointless; I don't even feel like listening to music.

Figure 11: Full doctor–patient dialogue in English (Page 3)

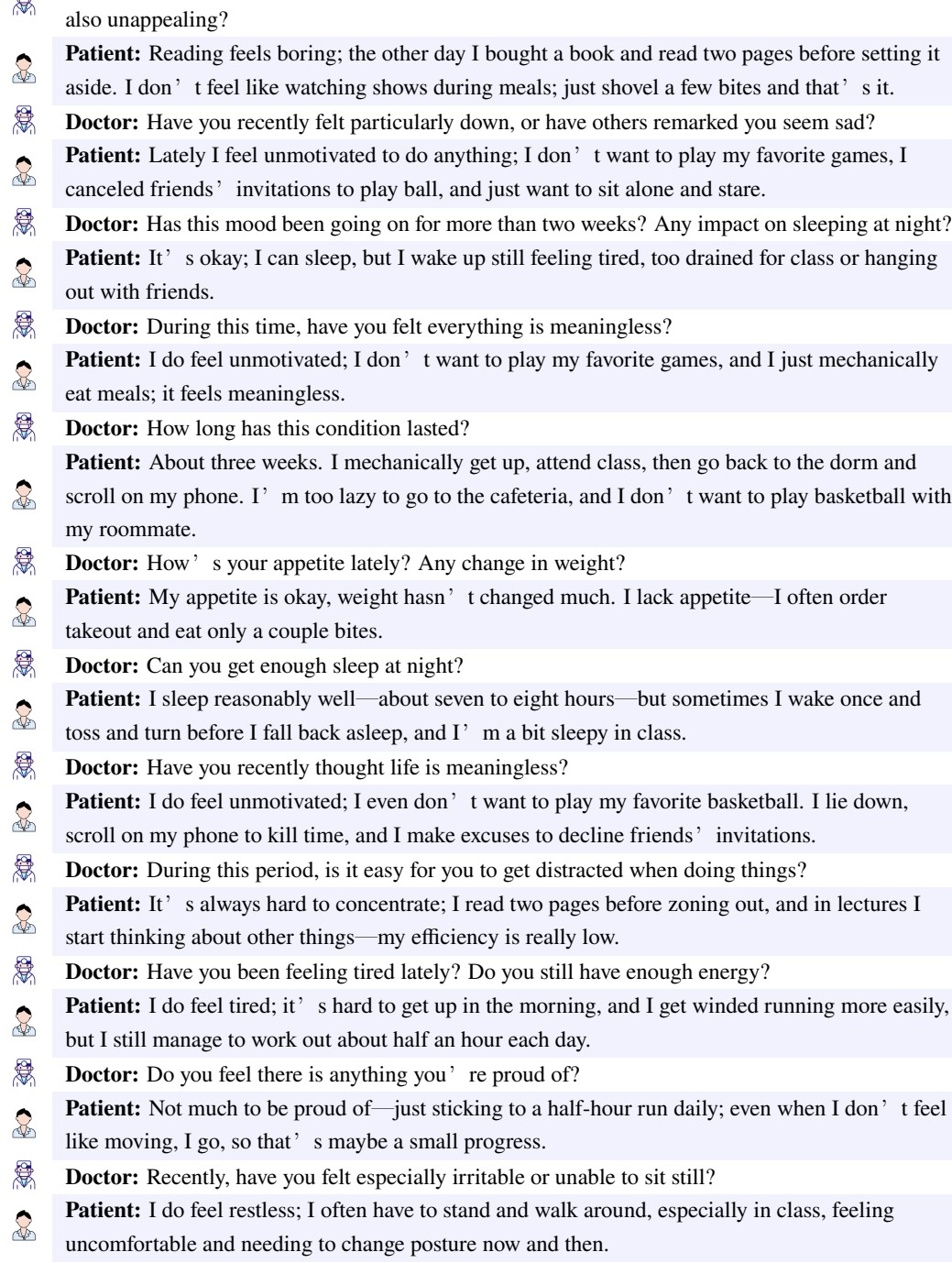

**Doctor:** Has this feeling lasted more than a week? Besides running and music, are other activities also unappealing?

**Patient:** Reading feels boring; the other day I bought a book and read two pages before setting it aside. I don't feel like watching shows during meals; just shovel a few bites and that's it.

**Doctor:** Have you recently felt particularly down, or have others remarked you seem sad?

**Patient:** Lately I feel unmotivated to do anything; I don't want to play my favorite games, I canceled friends' invitations to play ball, and just want to sit alone and stare.

**Doctor:** Has this mood been going on for more than two weeks? Any impact on sleeping at night?

**Patient:** It's okay; I can sleep, but I wake up still feeling tired, too drained for class or hanging out with friends.

**Doctor:** During this time, have you felt everything is meaningless?

**Patient:** I do feel unmotivated; I don't want to play my favorite games, and I just mechanically eat meals; it feels meaningless.

**Doctor:** How long has this condition lasted?

**Patient:** About three weeks. I mechanically get up, attend class, then go back to the dorm and scroll on my phone. I'm too lazy to go to the cafeteria, and I don't want to play basketball with my roommate.

**Doctor:** How's your appetite lately? Any change in weight?

**Patient:** My appetite is okay, weight hasn't changed much. I lack appetite—I often order takeout and eat only a couple bites.

**Doctor:** Can you get enough sleep at night?

**Patient:** I sleep reasonably well—about seven to eight hours—but sometimes I wake once and toss and turn before I fall back asleep, and I'm a bit sleepy in class.

**Doctor:** Have you recently thought life is meaningless?

**Patient:** I do feel unmotivated; I even don't want to play my favorite basketball. I lie down, scroll on my phone to kill time, and I make excuses to decline friends' invitations.

**Doctor:** During this period, is it easy for you to get distracted when doing things?

**Patient:** It's always hard to concentrate; I read two pages before zoning out, and in lectures I start thinking about other things—my efficiency is really low.

**Doctor:** Have you been feeling tired lately? Do you still have enough energy?

**Patient:** I do feel tired; it's hard to get up in the morning, and I get winded running more easily, but I still manage to work out about half an hour each day.

**Doctor:** Do you feel there is anything you're proud of?

**Patient:** Not much to be proud of—just sticking to a half-hour run daily; even when I don't feel like moving, I go, so that's maybe a small progress.

**Doctor:** Recently, have you felt especially irritable or unable to sit still?

**Patient:** I do feel restless; I often have to stand and walk around, especially in class, feeling uncomfortable and needing to change posture now and then.

Figure 12: Full doctor–patient dialogue in English (Page 4)

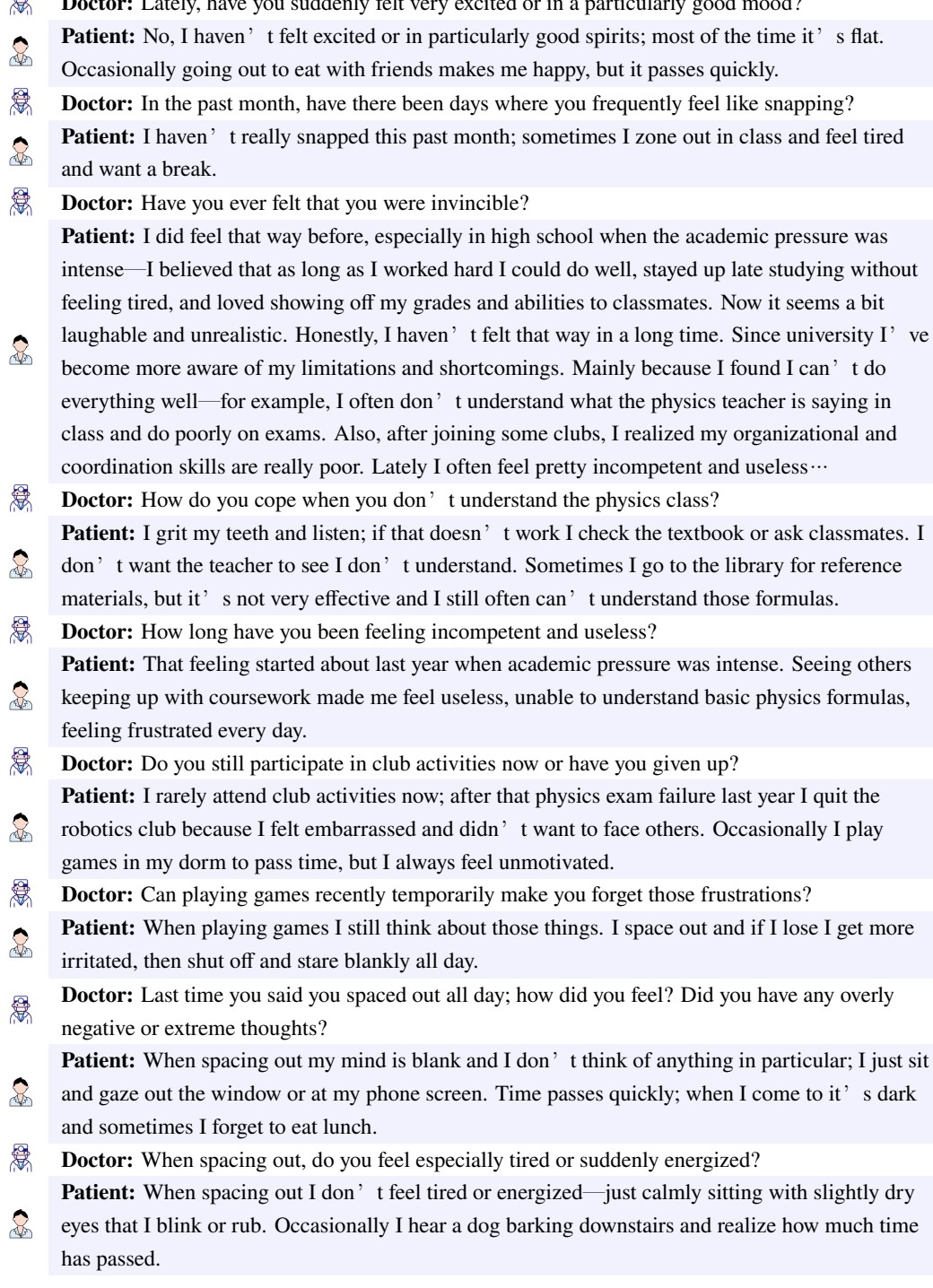

**Doctor:** Lately, have you suddenly felt very excited or in a particularly good mood?

**Patient:** No, I haven't felt excited or in particularly good spirits; most of the time it's flat. Occasionally going out to eat with friends makes me happy, but it passes quickly.

**Doctor:** In the past month, have there been days where you frequently feel like snapping?

**Patient:** I haven't really snapped this past month; sometimes I zone out in class and feel tired and want a break.

**Doctor:** Have you ever felt that you were invincible?

**Patient:** I did feel that way before, especially in high school when the academic pressure was intense—I believed that as long as I worked hard I could do well, stayed up late studying without feeling tired, and loved showing off my grades and abilities to classmates. Now it seems a bit laughable and unrealistic. Honestly, I haven't felt that way in a long time. Since university I've become more aware of my limitations and shortcomings. Mainly because I found I can't do everything well—for example, I often don't understand what the physics teacher is saying in class and do poorly on exams. Also, after joining some clubs, I realized my organizational and coordination skills are really poor. Lately I often feel pretty incompetent and useless···

**Doctor:** How do you cope when you don't understand the physics class?

**Patient:** I grit my teeth and listen; if that doesn't work I check the textbook or ask classmates. I don't want the teacher to see I don't understand. Sometimes I go to the library for reference materials, but it's not very effective and I still often can't understand those formulas.

**Doctor:** How long have you been feeling incompetent and useless?

**Patient:** That feeling started about last year when academic pressure was intense. Seeing others keeping up with coursework made me feel useless, unable to understand basic physics formulas, feeling frustrated every day.

**Doctor:** Do you still participate in club activities now or have you given up?

**Patient:** I rarely attend club activities now; after that physics exam failure last year I quit the robotics club because I felt embarrassed and didn't want to face others. Occasionally I play games in my dorm to pass time, but I always feel unmotivated.

**Doctor:** Can playing games recently temporarily make you forget those frustrations?

**Patient:** When playing games I still think about those things. I space out and if I lose I get more irritated, then shut off and stare blankly all day.

**Doctor:** Last time you said you spaced out all day; how did you feel? Did you have any overly negative or extreme thoughts?

**Patient:** When spacing out my mind is blank and I don't think of anything in particular; I just sit and gaze out the window or at my phone screen. Time passes quickly; when I come to it's dark and sometimes I forget to eat lunch.

**Doctor:** When spacing out, do you feel especially tired or suddenly energized?

**Patient:** When spacing out I don't feel tired or energized—just calmly sitting with slightly dry eyes that I blink or rub. Occasionally I hear a dog barking downstairs and realize how much time has passed.

Figure 13: Full doctor–patient dialogue in English (Page 5)

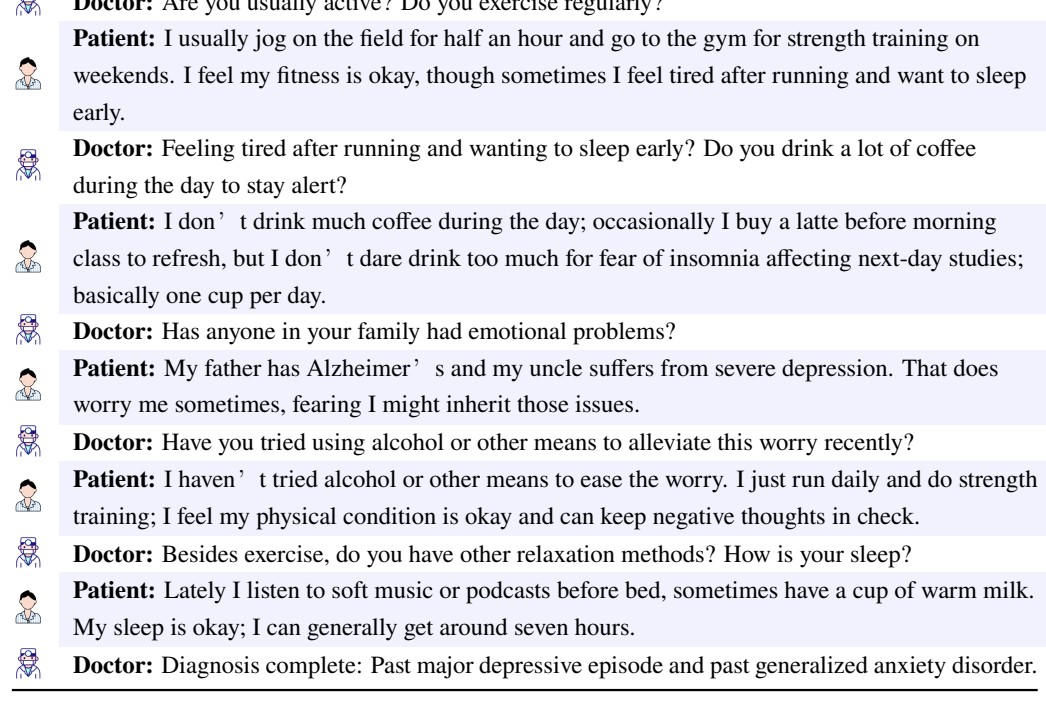

**Doctor:** Are you usually active? Do you exercise regularly?

**Patient:** I usually jog on the field for half an hour and go to the gym for strength training on weekends. I feel my fitness is okay, though sometimes I feel tired after running and want to sleep early.

**Doctor:** Feeling tired after running and wanting to sleep early? Do you drink a lot of coffee during the day to stay alert?

**Patient:** I don't drink much coffee during the day; occasionally I buy a latte before morning class to refresh, but I don't dare drink too much for fear of insomnia affecting next-day studies; basically one cup per day.

**Doctor:** Has anyone in your family had emotional problems?

**Patient:** My father has Alzheimer's and my uncle suffers from severe depression. That does worry me sometimes, fearing I might inherit those issues.

**Doctor:** Have you tried using alcohol or other means to alleviate this worry recently?

**Patient:** I haven't tried alcohol or other means to ease the worry. I just run daily and do strength training; I feel my physical condition is okay and can keep negative thoughts in check.

**Doctor:** Besides exercise, do you have other relaxation methods? How is your sleep?

**Patient:** Lately I listen to soft music or podcasts before bed, sometimes have a cup of warm milk. My sleep is okay; I can generally get around seven hours.

**Doctor:** Diagnosis complete: Past major depressive episode and past generalized anxiety disorder.

Figure 14: Full doctor–patient dialogue in English (Page 6)

