# OpenReview forum: "From Medical Records to Diagnostic Dialogues: A Clinical-Grounded Approach and Dataset for Psychiatric Comorbidity"
_ICLR.cc/2026/Conference — ICLR 2026 Poster_

### Official Review · Reviewer_tWSs · 2025-10-26

**Soundness:** 3
**Presentation:** 3
**Contribution:** 3
**Rating:** 8
**Confidence:** 3

**Summary:**

This paper presents a novel two-stage pipeline to address the challenge of diagnosing psychiatric comorbidity. First, the paper creates PsyCoProfile, a set of 502 synthetic, clinically-grounded Electronic Medical Records (EMRs) for patients with common comorbid disorders, derived from social media data. Second, the paper develops a multi-agent framework combining a hierarchical diagnostic state machine and the diagnosis context tree for diagnosis. Lastly, the paper introduces PsyCoTalk, the first large-scale dataset of 3,000 multi-turn diagnostic dialogues for psychiatric comorbidity, which has been validated by licensed psychiatrists for its clinical realism and diagnostic validity. This work provides a crucial resource for training and evaluating models capable of multi-disorder psychiatric screening.

**Strengths:**

1.The paper proposes PsyCoTalk, the first large-scale dialogue dataset specifically targeting psychiaatric comorbidity.

2.The paper's claims are substantiated by a rigorous and multi-faceted evaluation protocol. The results, particularly the high realism scores in an AB test against real-world dialogues, provide strong evidence for the dataset's clinical plausibility and validity.

3.The paper is well-presented, facilitating a strong understanding of the work's mechanics and contributions.

**Weaknesses:**

1.The model simplifies clinical symptoms to a basic "yes/no" answer, losing the nuance of real-world diagnosis where symptoms can be mild or subthreshold.

2.The framework relies on a manually-crafted state machine. While it can guarantee clinical accuracy, this approach is rigid and labor-intensive. Scaling the system to include more diseases would require significant expert effort to design and implement new state-machine modules, making the entire pipeline difficult to expand.

**Questions:**

1.Could the authors clarify the release plan for the assets created in this work? Specifically, will the full dataset of 502 synthetic EMRs (PsyCoProfile) be released alongside the 3,000 multi-turn diagnostic dialogues (PsyCoTalk)?

2.The paper demonstrates the quality of the dataset but does not establish a performance benchmark for models trained on it. To demonstrate the dataset's utility for downstream tasks, could the authors provide a performance report for a baseline model trained on PsyCoTalk for multi-disorder diagnosis? This would provide a crucial point of comparison for future works.

---

> ### Author Response · Authors · 2025-11-22
>
> 1. On the concern about simplifying four-level severity ratings to binary thresholds
>
> We thank the reviewer for raising this concern. We do not discard severity information; it is encoded in specific HDSM nodes and transitions, following DSM-5/SCID-5 requirements such as duration (A254 ≥ 1 week) and functional impairment (A23, whether symptoms affect school, work, or social functioning).
> We adopt binary threshold variables not because severity is removed, but because four-level ratings introduce instability that current LLMs cannot reliably manage. DSM-5/SCID-5 distinctions such as “below threshold vs. insufficient information” require nuanced clinical judgment, and early experiments showed that when LLMs were asked to maintain four-level severity, they frequently produced inconsistent or contradictory ratings, leading to diagnostic loops or invalid transitions. Using binary variables to represent only whether diagnostic thresholds are met avoids these failures and yields a more stable and auditable diagnostic process.
>
> 2. On the concern that the framework relies on manually constructed state machines
>
> We understand the concern about rigidity and expansion cost, but the HDSM is not hand-crafted from scratch. It is a direct programmatic encoding of SCID-5-RV, whose diagnostic branches and criteria already provide the full logic. Clinical experts do not redesign a new reasoning system; the state machine simply formalizes an existing, validated structure.
> The HDSM is also modular and reusable. Disorders such as depression, anxiety, ADHD, and bipolar share many symptom nodes (sleep, appetite, attention, somatic anxiety, etc.). Adding a new disorder typically requires only attaching disorder-specific branches and exclusion rules on top of shared modules, rather than building an entirely new graph. Thus, the marginal cost of extending to additional disorders is low.
> Finally, the “rigidity” is intentional. Encoding SCID-5-RV provides interpretability, auditability, and safety, ensuring the model operates within clinically validated boundaries rather than inventing ad-hoc diagnostic logic. The state machine is therefore a one-time infrastructure investment that enables scalable generation while preserving clinical fidelity.
>
> 3. On the question regarding the release plan for PsyCoTalk and PsyCoProfile
>
> We appreciate the reviewer’s attention to resource availability. Our release plan is as follows. We will publicly release the full PsyCoProfile dataset (502 synthetic EMRs) together with the 3,000 multi-turn diagnostic dialogues in PsyCoTalk. Both resources, along with all accompanying code, will be provided in a single repository so that the community can conduct research at different levels—using EMRs alone, dialogues alone, or the combined EMR + dialogue pipeline.
> During the anonymous review phase, full access instructions are already included in the supplementary materials. Upon acceptance, we will release PsyCoTalk and PsyCoProfile in a non-anonymized, fully open form, ensuring that this work serves not only as a methodological contribution but also as reusable data infrastructure for comorbidity research.
>
> 4. On the concern that the work lacks controlled evaluations of downstream model performance
>
> We acknowledge the reviewer’s concern that the current version does not include controlled evaluations, such as comparing models trained with and without PsyCoTalk. Our primary goal, however, is to introduce a systematic, clinically aligned framework for constructing comorbidity-aware psychiatric dialogue datasets (PsyCoProfile → PsyCoTalk) to address the severe scarcity of real comorbid interview data, rather than to demonstrate downstream model gains. This work provides the foundation for such future evaluation by releasing a reusable pipeline that transforms symptom cues into clinician-designed EMRs and multi-agent diagnostic interviews. PsyCoProfile, the first structured EMR resource for psychiatric comorbidity, and PsyCoTalk, a collection of 3,000 multi-turn diagnostic dialogues, fill a critical gap required for systematic comparison and reasoning studies. As stated in the Limitations section, more experiments and evaluations will be conducted in future work to quantify the dataset’s impact on model performance and diagnostic reasoning.

---

> > ### Comment · Reviewer_tWSs · 2025-11-26
> >
> > Thanks for your response. I will maintain my positive score.

---

### Official Review · Reviewer_ZBWt · 2025-10-30

**Soundness:** 2
**Presentation:** 3
**Contribution:** 3
**Rating:** 6
**Confidence:** 3

**Summary:**

This paper presents PsyCoTalk, a large-scale, clinically grounded dataset of 3,000 multi-turn psychiatric diagnostic dialogues generated from 502 synthetic electronic medical records.
The authors propose a multi-agent framework simulating doctor–patient interactions, guided by a Hierarchical Diagnostic State Machine  and a Diagnostic Context Tree based on DSM-5 structured interview standards.

The dataset focuses on psychiatric comorbidity, i.e., the co-occurrence of multiple mental disorders. Expert psychiatrists validated the data for realism and clinical validity, finding that PsyCoTalk conversations are close to real diagnostic interviews. The authors argue that this dataset will support the development of AI systems for multi-disease mental health reasoning and diagnostic support.

**Strengths:**

The pipeline integrates SCID-5 logic, diagnostic state transitions, and contextual reasoning, providing a strong medical foundation rarely seen in synthetic dialogue work.

Combines multi-agent dialogue simulation with structured EMR synthesis — a novel hybrid between symbolic reasoning and LLM-based text generation.

First dataset to explicitly address psychiatric comorbidity through structured, clinically grounded dialogues.

**Weaknesses:**

The synthetic medical records and the generated dialogues come from the same design logic. The “doctor” agent is judged against data that the system itself produced. This makes it hard to know whether the model is learning genuine clinical reasoning or just reproducing patterns it already encoded.

The diagnostic flow treats symptoms as mostly binary (“present” or “absent”), while real clinicians deal with uncertainty, partial symptoms, and differential diagnoses. The result may teach models to classify too confidently.

The dataset only covers a small set of disorder combinations (mainly depression, anxiety, bipolar, ADHD). Other common co-occurring conditions like PTSD or substance use are missing, so “comorbidity” here is still narrow.

The evaluation mainly compares PsyCoTalk to versions of itself (with or without the diagnostic controller). There’s no test showing whether this data helps models perform better on external or real-world benchmarks.

**Questions:**

N/A

---

> ### Author Response · Authors · 2025-11-22
>
> 1. On the concern that synthetic EMRs and dialogues share the same logic
>
> We appreciate the concern. Two clarifications address it directly.
> First, PsyCoProfile and PsyCoTalk do not share the same generation logic. EMRs are produced from social-media symptom cues using psychiatrist-designed templates to ensure clinically complete case narratives. This stage does not involve SCID-5-RV reasoning. The dialogue stage introduces the SCID-5-RV–compiled hierarchical diagnostic state machine together with the multi-agent setup: the tool agent drives diagnostic transitions, the doctor agent generates context-aware questions, and the patient agent answers according to its EMR. EMRs therefore provide clinical content, whereas dialogues implement diagnostic reasoning. These are two different layers rather than a duplicated logic.
> Second, evaluation is external and independent of the system. We do not rely on internal agent signals. Quality is assessed through statistical comparison with 1,731 real clinical consultations and through double-blind ratings by five licensed psychiatrists who evaluate fifty dialogues and perform AB-tests mixing PsyCoTalk with real and synthetic data. Evaluation is therefore carried out by independent clinicians and real-world baselines rather than by the system itself.
>
> 2. On the concern about simplifying four-level severity ratings to binary thresholds
>
> We thank the reviewer for raising this concern. We do not discard severity information; it is encoded in specific HDSM nodes and transitions, following DSM-5/SCID-5 requirements such as duration (A254 ≥ 1 week) and functional impairment (A23, whether symptoms affect school, work, or social functioning).
> We adopt binary threshold variables not because severity is removed, but because four-level ratings introduce instability that current LLMs cannot reliably manage. DSM-5/SCID-5 distinctions such as “below threshold vs. insufficient information” require nuanced clinical judgment, and early experiments showed that when LLMs were asked to maintain four-level severity, they frequently produced inconsistent or contradictory ratings, leading to diagnostic loops or invalid transitions. Using binary variables to represent only whether diagnostic thresholds are met avoids these failures and yields a more stable and auditable diagnostic process.
>
> 3. On the concern that the comorbidity coverage is limited to a small set of disorders
>
> We agree that the current release focuses on a subset of disorders (depression, anxiety, bipolar disorder, ADHD). This choice is driven not by conceptual narrowness, but by the practical difficulty of obtaining clinically reliable comorbidity data. Our original pool includes seven disorders and over 180 possible comorbidity combinations. To ensure clinical validity, we apply strict selection criteria: we retain only users with ≥10 symptom-related posts and ≥20 distinct symptom types, and we filter symptom–diagnosis mappings using DSM-5 consistency checks (Section 3). A comorbidity combination is included only when its sample size exceeds 50, so that each retained category contains sufficiently rich and stable symptom information for meaningful EMR construction and dialogue generation.
> This principled filtering ensures that every retained comorbidity group has high-quality, clinically plausible EMRs rather than sparse, noisy cases. Although the final set does not cover all possible disorders such as PTSD or substance use, it constitutes the first structured EMR resource specifically designed for psychiatric comorbidity, addressing a gap where real clinical data are nearly unobtainable. The framework itself is not limited to these four disorders and can be extended to additional conditions in future versions as reliable source data become available.
>
> 4. On the concern that the work lacks controlled evaluations of downstream model performance
>
> We acknowledge the reviewer’s concern that the current version does not include controlled evaluations, such as comparing models trained with and without PsyCoTalk. Our primary goal, however, is to introduce a systematic, clinically aligned framework for constructing comorbidity-aware psychiatric dialogue datasets (PsyCoProfile → PsyCoTalk) to address the severe scarcity of real comorbid interview data, rather than to demonstrate downstream model gains. This work provides the foundation for such future evaluation by releasing a reusable pipeline that transforms symptom cues into clinician-designed EMRs and multi-agent diagnostic interviews. PsyCoProfile, the first structured EMR resource for psychiatric comorbidity, and PsyCoTalk, a collection of 3,000 multi-turn diagnostic dialogues, fill a critical gap required for systematic comparison and reasoning studies. As stated in the Limitations section, more experiments and evaluations will be conducted in future work to quantify the dataset’s impact on model performance and diagnostic reasoning.

---

> > ### Comment · Reviewer_ZBWt · 2025-11-27
> >
> > thanks for your response, I believe my score is fair for this work, I will keep OA 6 unless interactions between authors and other reviewers change my idea.

---

### Official Review · Reviewer_SXir · 2025-10-30

**Soundness:** 3
**Presentation:** 2
**Contribution:** 3
**Rating:** 4
**Confidence:** 4

**Summary:**

This paper presents PsyCoTalk, a clinically grounded dataset for psychiatric comorbidity diagnosis. It introduces a two-stage pipeline:
(1) PsyCoProfile, which constructs 502 synthetic electronic medical records (EMRs) for patients with multiple co-occurring disorders (MDD, AD, BD, ADHD), generated from social media posts; and
(2) PsyCoTalk, which uses a multi-agent framework guided by SCID-5-RV (a DSM-5 diagnostic interview standard) to produce 3,000 multi-turn diagnostic dialogues.
The dataset is validated by psychiatrists and claimed to be the first resource that supports comorbid diagnostic reasoning in simulated doctor–patient interactions.

**Strengths:**

(1)Novel focus on psychiatric comorbidity: Unlike prior mental disorder datasets that focus on single disorders, this work explicitly targets psychiatric comorbidity, which is a clinically important setting.
(2)Multi-agent framework: Integrating doctor, patient, and tool agents under a hierarchical diagnostic state machine (HDSM) is interpretable.
(3)Psychiatrist validation: Involvement of licensed psychiatrists adds credibility to the dataset’s linguistic and diagnostic realism.

**Weaknesses:**

(1)Data effectiveness: Since all EMRs and dialogues are synthetic, derived from social media posts and LLM-based generation, can the dataset truly reflect authentic doctor–patient interactions? Do the linguistic patterns or emotional tone in these generated dialogues capture the depth and subtlety of real psychiatric interviews? Without any real clinical data for grounding or comparison, how credible is the claim of “clinical realism”?
(2)Simplified symptom representation: By reducing the SCID-5’s original four-point symptom scale to binary “present/absent” labels, does the dataset lose essential diagnostic nuance and symptom severity? How might this simplification affect the interpretability and downstream clinical reliability of models trained on such data?
(3)Lack of Inter-Disorder Relationship Modeling: If “comorbidity” here merely refers to co-labeled samples without modeling relationships or dependencies between disorders, can it truly represent comorbid diagnostic reasoning? Within the dialogues, do co-occurring disorders interact dynamically, or are they treated as independent diagnostic categories? To what extent does the dialogue structure capture overlapping or interacting symptoms?
(4)Evaluation limitations: Given that only 50 dialogues were rated by five psychiatrists, is this sample sufficient to demonstrate the dataset’s overall reliability, linguistic realism, and diagnostic accuracy? How representative are those 50 evaluated samples compared to the entire corpus of 3,000 dialogues, and how consistent are the expert ratings?

**Questions:**

Please check the weakness part

---

> ### Author Response · Authors · 2025-11-22
>
> 1. On “data validity / clinical realism”
>
> We acknowledge the concern about whether synthetic data can reflect real psychiatric encounters. Our realism claims are supported by multi-level evaluation using real clinical data and expert review. At the EMR level, Reddit posts are used only as symptom cues; psychiatrist-designed templates generate structured EMRs, which we compare with ~1,000 de-identified clinical EMRs. Disease spectrum, sex, family history, and age distributions closely match real data, and eight psychiatrists reviewed sampled EMRs to refine generation rules and ensure medical validity. At the dialogue level, PsyCoTalk is benchmarked against 1,731 real clinical consultations. Turn length, utterance length, entropy, hapax ratio, and semantic diversity show PsyCoTalk aligns most closely with real dialogues rather than template-like patterns (see Table 2 and Appendix A.2 Table 5). We further conduct subjective evaluation: fifty dialogues rated double-blind by five licensed psychiatrists achieve high realism scores and rank second only to real dialogues in an AB test, outperforming D₄ and MDD-5k. Together, these real-data comparisons and double-blind expert ratings demonstrate strong clinical realism. We do not claim synthetic data replace real records, but our multi-layer evaluation provides credible evidence of realism, since real psychiatric data acquisition exhibits stringent ethical and safety concerns, even for research purpose.
>
>
> 2. On “symptom simplification: 4-level scale → binary labels”
>
> We thank the reviewer for raising this concern. We do not discard severity information; it is encoded in specific HDSM nodes and transitions, following DSM-5/SCID-5 requirements such as duration (A254 ≥ 1 week) and functional impairment (A23, whether symptoms affect school, work, or social functioning).
> We adopt binary threshold variables not because severity is removed, but because four-level ratings introduce instability that current LLMs cannot reliably manage. DSM-5/SCID-5 distinctions such as “below threshold vs. insufficient information” require nuanced clinical judgment, and early experiments showed that when LLMs were asked to maintain four-level severity, they frequently produced inconsistent or contradictory ratings, leading to diagnostic loops or invalid transitions. Using binary variables to represent only whether diagnostic thresholds are met avoids these failures and yields a more stable and auditable diagnostic process.
>
>
> 3. On “whether comorbidity reasoning is truly modeled”
>
> We agree that comorbidity must reflect structured diagnostic interaction rather than independent multi-label assignment. Our HDSM encodes cross-disorder dependencies directly from SCID-5-RV, which already specifies clinically validated differential-diagnosis logic. Distinctions such as MDD vs. GAD (duration, impairment), bipolar vs. unipolar depression ((hypo)mania), and substance-induced vs. primary mood symptoms are implemented as cross-node transitions and priority rules. This yields unified, interpretable joint reasoning on a single state machine rather than parallel labels.
> SCID-5-RV’s diagnostic structure has been widely validated [1][2][3], and our modeling inherits this clinically established framework rather than introducing new assumptions.
>
>
> 4. On “whether 50 dialogues / five psychiatrists are sufficient”
>
> We appreciate the concern. The 50 samples are randomly drawn from 3,000 dialogues, and their diagnostic and demographic distributions closely match the full dataset (depression 0.35 vs. 0.38, anxiety 0.32 vs. 0.31, ADHD 0.21 vs. 0.19, bipolar 0.12 vs. 0.12; female ratio 0.53 vs. 0.50). Each dialogue is rated by all five psychiatrists across six dimensions. Inter-rater consistency is high, with scores above 0.8 for all dimensions (mean 0.836), indicating stable and reliable judgments.
> Our evaluation scale is comparable to, or more rigorous than, prior top-tier work. We evaluate 50 dialogues (1.67% of 3,000), a proportional rate equal to or higher than comparable studies. KokoroChat [4] evaluates 100 samples (\~0.15%) with 5 counselors; MDD-5k [5] evaluates 100 (\~2%) with 5 clinicians; Can AI Relate [6] evaluates 50 (\~0.4%) with 2 psychologists; and MentaLLaMA [7] evaluates 200 (\~0.2%) with 3 psychology researchers. LLM-Clinical [8] conducts clinician evaluation on 140 questions with 9 clinicians, and DeepSeek-CDS [9] evaluates 125 clinical vignettes using 5-point Likert ratings with confirmed inter-rater reliability.
>
>
>
> References
>
> [1] Mohammadkhani et al. (2020). SCID-Persian-Validation.
>
> [2] Brodey et al. (2018). SAGE-SR Validation.
>
> [3] Shankman et al. (2018). SCID-Dimensional-Severity.
>
> [4] Qi et al. (2025). KokoroChat (ACL).
>
> [5] Yin et al. (2025). MDD-5k (AAAI).
>
> [6] Gabriel et al. (2024). Can AI Relate? (EMNLP).
>
> [7] Yang et al. (2024). MentaLLaMA (WWW).
>
> [8] Singhal et al. (2023). LLM-Clinical (Nature).
>
> [9] Sandmann et al. (2025). DeepSeek-CDS (Nature Medicine).

---

### Official Review · Reviewer_rh5N · 2025-11-02

**Soundness:** 3
**Presentation:** 3
**Contribution:** 3
**Rating:** 6
**Confidence:** 3

**Summary:**

This paper introduces a clinically grounded framework and dataset for diagnostic dialogues in the context of psychiatric comorbidity, aiming to bridge the gap between static medical records and interactive diagnostic conversations. The authors derive structured psychiatric dialogue data from clinical records and design a dialogue system that models comorbidity-aware diagnostic reasoning. The dataset includes multi-turn dialogues reflecting realistic clinician–patient interactions and overlapping psychiatric conditions, a setting that is significantly underrepresented in existing diagnostic dialogue research.

**Strengths:**

- Clinically meaningful and socially impactful contribution. Psychiatric comorbidities are extremely common in real clinical practice, yet rarely addressed in diagnostic dialogue datasets. The paper fills an important gap by designing dialogues that reflect comorbidity patterns, symptom overlap, and ambiguity, which are critical challenges for mental health assessments.
- Clear dataset design and annotation strategy. The paper provides a transparent methodology for constructing dialogues, annotating comorbid conditions, and representing psychiatric symptomatology in a conversation-friendly format. The schema and examples help convey the complexity handled by the dataset.
- Well-written and structured. The paper is generally clear and does a good job situating itself relative to existing medical dialogue datasets, noting that most lack comorbidity or clinically grounded language patterns.

**Weaknesses:**

- Lack of rigorous evaluation. The paper does not present systematic evaluation of the dataset’s usefulness beyond illustrative examples. No comparisons or user studies (e.g., models trained with vs. without this dataset) are provided to show the dataset’s impact on model performance or clinical reasoning.
- Limited novelty in methodology. The main novelty is the dataset’s domain focus. The data transformation pipeline is not sufficiently innovative or thoroughly justified for ICLR.
- The dataset description could better highlight diversity: e.g., demographic representation, cultural/linguistic bias, and how psychiatric conditions with culturally variant presentations were handled.

**Questions:**

- How do you envision safe and appropriate use of this dataset, given the risk of non-professional misuse of psychiatric diagnostic tools?
- Were any psychiatrists, clinical psychologists, or licensed clinicians involved in reviewing the dataset or annotation guidelines? If so, please include details.

**Details Of Ethics Concerns:**

The dataset is derived from psychiatric medical records, which contains sensitive health data. The paper does not sufficiently clarify the de-identification process, data governance, or whether clinical/ethical oversight was obtained. In addition, psychiatric diagnostic conversations pose potential risks if used without appropriate guardrails or clinician supervision. An ethics review may help ensure responsible data release and usage guidelines.

---

> ### Author Response · Authors · 2025-11-22
>
> 1. On the concern “lack of rigorous evaluation” regarding model performance or clinical reasoning
>
> We acknowledge the reviewer’s concern that the current version does not include controlled evaluations, such as comparing models trained with and without PsyCoTalk. Our primary goal, however, is to introduce a systematic, clinically aligned framework for constructing comorbidity-aware psychiatric dialogue datasets (PsyCoProfile → PsyCoTalk) to address the severe scarcity of real comorbid interview data, rather than to demonstrate downstream model gains. This work provides the foundation for such future evaluation by releasing a reusable pipeline that transforms symptom cues into clinician-designed EMRs and multi-agent diagnostic interviews. PsyCoProfile, the first structured EMR resource for psychiatric comorbidity, and PsyCoTalk, a collection of 3,000 multi-turn diagnostic dialogues, fill a critical gap required for systematic comparison and reasoning studies. As stated in the Limitations section, more experiments and evaluations will be conducted in future work to quantify the dataset’s impact on model performance and diagnostic reasoning.
>
> 2.  On “limited methodological novelty”
>
> While the reviewer characterizes our approach as a “data transformation pipeline,” our contribution is substantially more than that. This work introduces the first clinically grounded generation framework for comorbidity-aware psychiatric diagnostic dialogues, combining (1) clinician-co-designed structured EMR templates (rather than rewriting social-media text), (2) a SCID-5-RV–compiled hierarchical diagnostic state machine encoding 100+ clinically validated decision nodes, and (3) a multi-agent reasoning setup that couples symbolic SCID logic with LLM-based doctor–patient interaction. This design yields clinically coherent, diagnostically traceable interviews, providing methodological innovation that goes well beyond domain-specific data conversion and establishes a reusable template for future clinical-reasoning research.
>
> 3. On “data diversity and cultural bias”
>
> We acknowledge that Reddit-derived symptom cues may introduce demographic or cultural skew. Our pipeline, however, is inherently transferable: it is not tied to any language, culture, or platform, and the same EMR template and HDSM can generate localized datasets wherever structured mental-health records or dialogues exist. All dialogues follow SCID-5-RV, an internationally validated interview used across cultures, providing a stable clinical backbone. The initial release uses Reddit cues and Chinese outputs due to expert-evaluation constraints, and we plan multilingual and multicultural extensions while encouraging other groups to adapt the framework to local clinical data to reduce cultural bias.
>
> 4. On “safety and compliant use boundaries”
>
> We agree with the reviewer’s safety concerns. As stated in the Ethics Statement, the dataset is for research only, designed for multi-disorder screening and dialogue modelling, and is not a diagnostic tool or a source of treatment advice. To prevent misuse: (1) Research-only restriction: all EMRs and dialogues are synthetic and must not be used for individual clinical decisions or interpreted as clinical guidance. (2) Data-usage agreement: access requires signing an agreement that prohibits clinical deployment, real-user psychological services, and any re-identification attempts. Any real-world mental-health use would require formal ethics review and clinician oversight.
>
> 5. On “whether clinical professionals participated”
>
> Yes. Licensed psychiatrists were involved at multiple stages: 1) EMR template definition and EMR construction. Psychiatrists co-designed the structured EMR template (chief complaint, family history, etc.). During EMR generation, we randomly sampled cases for expert review by eight psychiatrists; their feedback guided revisions to ensure medical validity. 2) Final human evaluation of PsyCoTalk. As stated in the Evaluation section (p.8), each dialogue was scored by all five psychiatrists across six dimensions (professionalism, communication quality, fluency, realism, etc.) and compared with real clinical dialogues through an AB-test.

---

### Author Response · Authors · 2025-11-22
**Ethical Considerations & Data Governance**

Because several reviewers marked an ethics flag, we provide a consolidated clarification of our data-governance procedures.

1. Social-media data originate from the PsySym dataset under a formal data-use agreement and were strictly de-identified (removal of usernames, timestamps, and locations). The original dataset underlying PsySym was approved by IRB I2022158P and complies with all relevant privacy regulations.

2. PsyCoProfile is generated entirely from these de-identified symptom cues, producing fully synthetic EMRs whose personal histories and life events are LLM-generated and not traceable to real individuals.

3. SCID-5 is used solely as a diagnostic framework. Human involvement is limited to evaluation by five licensed psychiatrists; no real patient data are used. All experts provided informed consent and were compensated according to responsible-research guidelines.

---

### Author Response · Authors · 2025-12-01

We thank reviewers for their appreciation and constructive suggestions as well as AC's help. Below is a brief summary for AC after rebuttal.
## Summary of contributions
The paper proposes a clinically grounded, two-stage framework for psychiatric comorbidity diagnosis and releases two resources: PsyCoProfile, 502 synthetic but clinically structured EMRs for patients with common comorbid combinations (MDD, anxiety, bipolar disorder, ADHD), created from de-identified PsySym symptom cues using psychiatrist-designed templates; and PsyCoTalk, 3,000 multi-turn diagnostic dialogues generated via a multi-agent framework that couples a SCID-5-RV–compiled Hierarchical Diagnostic State Machine (HDSM) and Diagnostic Context Tree with LLM-based doctor, patient, and tool agents, yielding comorbidity-aware, diagnostically traceable interviews. Together, these resources target a setting largely missing in current work: multi-disorder psychiatric reasoning in interactive dialogue.
## Clarifications on methodology, realism, and comorbidity
Several reviews question whether the work is just a data transformation pipeline and whether synthetic data reflect clinical encounters:
- Clinical realism is supported by multi-level comparison to real data and expert review. EMRs are matched against ~1,000 de-identified clinical EMRs for disorder spectrum and demographic distributions. Eight psychiatrists reviewed samples to refine generation rules. PsyCoTalk is compared against 1,731 real consultations and synthetic baselines; discourse statistics (turn length, lexical diversity, etc.) place PsyCoTalk closest to real dialogues rather than templated ones. In a double-blind study, five licensed psychiatrists rate 50 randomly sampled dialogues (diagnostic distributions closely match the full set) and run AB tests mixing real, PsyCoTalk, and other datasets; PsyCoTalk ranks second only to real dialogues.
- On comorbidity reasoning, comorbidity is not just multi-label tagging. The HDSM programmatically encodes SCID-5-RV cross-disorder dependencies and exclusion rules (e.g., MDD vs GAD, bipolar vs unipolar depression) on a single state machine, so overlapping symptoms and differential diagnosis are explicitly modeled during dialogue, rather than treated as independent labels. SCID-5-RV’s diagnostic structure is widely validated in prior work.
- On binary vs four-level symptom scales, the work does not discard severity. Duration and impairment thresholds are captured in specific HDSM nodes and transitions following DSM-5/SCID-5. Binary variables are used only to indicate whether diagnostic thresholds are met. Early experiments with four-level SCID scales led current LLMs to produce inconsistent severity ratings and invalid state transitions; binary thresholds were therefore adopted to keep the state machine stable and auditable while staying aligned with DSM-5 criteria.
- In addition, in response to concerns about whether 50 dialogues rated by five psychiatrists are sufficient, we show that the 50 samples match the full dataset in diagnostic and demographic distributions (e.g., depression 0.35 vs. 0.38, anxiety 0.32 vs. 0.31, ADHD 0.21 vs. 0.19, bipolar 0.12 vs. 0.12; female ratio 0.53 vs. 0.50), achieve high inter-rater consistency (all dimensions >0.8, mean 0.836), and correspond to 1.67% of PsyCoTalk, a proportion comparable to or more rigorous than six recent top-tier mental-health dialogue and LLM studies (details and specific references are provided in the rebuttal).

The current release covers four disorders due to strict quality filters; each retained comorbidity category has enough data for reliable EMRs. The modular framework can extend to more conditions as reliable data become available.
## Positioning w.r.t. evaluation
Several reviewers note that the paper does not yet provide controlled downstream benchmarks (e.g., models trained with/without PsyCoTalk). We explicitly position the work as infrastructure: a clinically aligned pipeline and dataset family for comorbidity-aware dialogues in a domain where true clinical interviews are extremely difficult to obtain. In future work, we plan to add controlled downstream benchmarks on top of this reusable, well-validated resource and methodology.
## Ethics and data governance
Social-media data come from the PsySym dataset under a formal data agreement and were strictly de-identified (removal of usernames, timestamps, locations); the original PsySym dataset was approved by IRB I2022158P and complies with privacy regulations. PsyCoProfile is generated entirely from these de-identified symptom texts, producing fully synthetic EMRs whose personal histories and life events are LLM-generated and not traceable to real individuals. SCID-5 is used solely as a diagnostic framework. The only human involvement is expert evaluation by five licensed psychiatrists; no real patient data are used. These experts provided informed consent and were compensated according to responsible-research guidelines.

---

### Meta-Review · Area_Chair_bt8Q · 2026-01-08

**Summary:**

The paper's contribution is mainly on the dataset side, and such medical records for psychiatric diagnostic conversations are valuable. The authors develop a two-stage framework generating two new resources: (1) synthetic but clinically structured EMRs for patients with common comorbid combinations and (2) multi-turn diagnostic dialogues created via multi-agent simulation. There are still some concerns that synthetic data may not truly reflect clinical realism and whether comorbidity reasoning is properly modeled rather than just multi-label classification. Also, the dataset covers only four disorder combinations and lacks downstream benchmarks showing whether models actually improve with this data. Symptom representation being simplified to binary scales, though authors clarified that severity is encoded through diagnostic thresholds. Generally the dataset is worth to be an accepted paper at ICLR and resources/experimental results will contribute to the community. The rebuttals are expected to be incorporated into the final version.

**Reviewer Concerns:**

There are still some concerns that synthetic data may not truly reflect clinical realism and whether comorbidity reasoning is properly modeled rather than just multi-label classification. Also, the dataset covers only four disorder combinations (limiting scope) and lacks downstream benchmarks showing whether models actually improve with this data. Symptom representation being simplified to binary scales, though authors clarified that severity is encoded through diagnostic thresholds.

**Reviewer Scores:**

All reviewers confirmed checking the rebuttals and get feedback. The paper would likely move from mixed (4, 6, 6, 8) to more positive (6, 6, 7, 8), with two solid accepts and two borderline accepts, making it a clear acceptance with poster recommendation.

---

### Decision · Program_Chairs · 2026-01-26

Accept (Poster)